# Disorder-induced enhancement of lithium-ion transport in solid-state electrolytes

Zhimin Chen[1], Tao Du ⓘ [1,2] ✉, N. M. Anoop Krishnan ⓘ [3], Yuanzheng Yue ⓘ [1] & Morten M. Smedskjaer ⓘ [1] ✉

Enhancing the ion conduction in solid electrolytes is critically important for the development of high-performance all-solid-state lithium-ion batteries (LIBs). Lithium thiophosphates are among the most promising solid electrolytes, as they exhibit superionic conductivity at room temperature. However, the lack of comprehensive understanding of their ion conduction mechanism, especially the effect of structural disorder on ionic conductivity, is a longstanding problem that limits further innovations in all-solid-state LIBs. Here, we address this challenge by establishing and employing a deep learning potential to simulate $Li_3PS_4$ electrolyte systems with varying levels of disorder. The results show that disorder-driven diffusion dynamics significantly enhances the room-temperature conductivity. We further establish bridges between dynamical characteristics, local structural features, and atomic rearrangements by applying a machine learning-based structure fingerprint termed "softness". This metric allows the classification of the disorder-induced "soft" hopping lithium ions. Our findings offer insights into ion conduction mechanisms in complex disordered structures, thereby contributing to the development of superior solid-state electrolytes for LIBs.

Lithium-ion batteries (LIBs) have revolutionized portable electronics and play an increasingly important role in electric vehicles and grid energy storage due to their high energy density and long cycle life[1–3]. However, as demands for higher energy density, enhanced safety, and faster charging continue to grow, traditional LIBs are rapidly approaching their performance limits[4–6], underscoring the urgent need to explore new frontiers in battery technology. One of the most promising avenues is the development of solid-state batteries, in which the liquid electrolyte is replaced with a solid electrolyte, enabling safer, more efficient, and extended lifespan in energy storage solutions. Unlike their liquid counterparts, solid electrolytes are non-flammable and intrinsically safer[7], mitigating the risk of thermal runaway events[8,9], leakage[10], and chemical instability[11]. Glassy solid electrolytes are interesting candidates for solid-state batteries, considering their various advantages over the crystalline counterparts such as isotropic ion conduction[12], lack of grain boundaries[13], and ease of industrial

processing[14]. Among these solid-state electrolytes, lithium thiophosphate based glasses and glass-ceramics are especially promising due to their high ionic conductivity[15–17], minimal ion transfer resistance at the electrode interface, and cost-effective processing[18].

The mechanism of ion conduction within solid-state electrolytes is fundamentally different from that of liquid electrolytes. In liquid electrolytes, ions move through the liquid medium and electrons are separated[19], while in solid-state electrolytes, ions diffuse along favorable migration pathways in crystals[7] or navigate a disordered structural landscape[20]. Understanding the intricacies of ion conduction in solid-state electrolytes is critical for optimizing battery performance. The mechanisms of ion conduction in crystalline solid electrolytes include vacancy-assisted migration, interstitial diffusion, and even tunneling[7]. However, the mechanisms for the ion conduction in disordered or glassy solid electrolytes have yet to be established[7,19,21], primarily because of the inherently irregular nature of the energy landscape in

[1]Department of Chemistry and Bioscience, Aalborg University, Aalborg East, Denmark. [2]Department of Applied Physics, The Hong Kong Polytechnic University, Kowloon, Hong Kong, China. [3]Department of Civil Engineering, Indian Institute of Technology Delhi, New Delhi, India. ✉e-mail: tao.du@polyu.edu.hk; mos@bio.aau.dk

these materials[20]. Furthermore, the lack of a well-defined crystal lattice makes it challenging to predict and control ion pathways accurately. Additionally, the presence of defects and disorder-induced structural heterogeneity further complicate the understanding of ion conduction in these materials.

Beyond glassy solid electrolytes with disordered structures, glass-ceramics are an important class of inorganic materials composed of crystals dispersed in a glass matrix[22]. This dual nature can combine the advantages of glass with the unique properties of the crystalline phase, as glass-ceramic electrolytes often outperform both pure glass and fully crystalline electrolytes in terms of ionic conduction[16,23–27]. However, their complicated structures make revealing their ion transport mechanisms more challenging. Additionally, analogous to grain boundaries in polycrystalline materials, interfaces are present between the crystalline and glassy phases, which may be termed complexions, as they are thermodynamically stable and physically distinct from the neighboring phases[28–30]. These interfaces in solid electrolytes have been found to play a crucial role in inhibiting lithium dendrite growth[30] and enhancing fracture resistance[31]. However, the definition of these interfaces is in some cases ambiguous[32] or defined to serve specific properties[31], and the understanding of ion dynamics at the interfaces requires further investigations.

The use of large-scale molecular dynamics (MD) simulations enables capturing the long-term (up to hundreds of nanoseconds) dynamics of hopping ions. However, high accuracy and efficiency potentials are required for simulating processes like the glass transition and phosphorus-sulfur (P-S) bond breaking in the important family of lithium phosphorus sulfide (LiPS) systems. It is worth noting that the recent classical potential proposed by Ariga et al.[33] has certain limitations, particularly with respect to accounting for bond breaking and reactions (e.g., the interconversion between $PS_4^{3-}$, $P_2S_6^{4-}$, and $P_2S_7^{4-}$ within the $Li_2S$-$P_2S_5$ glasses). Similarly, the classical potential proposed by Kim et al.[32] not only prohibits bond breaking but is also limited to $\gamma$-$Li_3PS_4$. To address this, we here train a machine learning-based interatomic potential (MLIP)[34] based on ab initio molecular dynamics (AIMD) training data. As shown herein, this new potential allows for the simulation of both crystalline and glassy forms of LiPS with an accuracy comparable with AIMD simulations but at a much lower computational cost.

With the objective to unravel the structural origins of disordered-induced acceleration of lithium-ion migration, we use the MLIP to construct glassy $Li_3PS_4$ electrolytes, as well as ordered $\beta$-$Li_3PS_4$ (*Pnma*) and partially crystalline $Li_3PS_4$ glass-ceramics (i.e., three systems with varying degree of disorder), and quantify structural descriptors for order and disorder. Further, we compare the homogeneous dynamics (mean-squared displacement, van Hove correlation functions) and heterogeneous dynamics (non-Gaussian statistics) across the systems. We then focus on the partially crystalline $Li_3PS_4$ electrolyte, quantifying the dynamic distinctions between its internal ordered and disordered phases, including disordered interfaces. Finally, we employ the classification-based structure fingerprint termed softness developed by Cubuk et al. (see *Methods* section)[35–37] to identify disorder-induced soft hopping ions. Taken as a whole, the present investigation of hopping ions' conduction mechanisms in solid-state electrolytes reveals the origin of disorder-driven fast transport of lithium ions. Notably, the combined approach involving molecular dynamics and machine learning is broadly applicable to studying other types of solid-state ion transport systems.

## Results

### Machine learning interatomic potential

To strike a balance between accuracy and computational efficiency, we develop a MLIP for the LiPS system. The MLIP is trained using trajectories obtained from density functional theory (DFT) level AIMD simulations as the dataset (see *Methods* section for details). The

accuracy of the MLIP in reproducing structural information obtained from AIMD is shown in Fig. 1a. The DeePMD based potential demonstrates an almost DFT-level accuracy in reproducing short-range structural features, closely matching the first and second coordination shell in the pair distribution function $g(r)$. Moreover, compared to MD simulations using a classical potential[33], which also cannot capture bond breaking and reaction events, the present MLIP-based simulations consistently exhibit a better agreement with the AIMD results.

We also compare the lattice parameters, activation energy, and ionic conductivity obtained from MLIP simulations with experimental values and various DFT calculation values. The detailed comparison can be found in Supplementary Table S1. For $\beta$-$Li_3PS_4$, the activation energy of impedance spectroscopy is relatively low, typically ranging between 0.30 and 0.50 eV[38,39]. Nuclear magnetic resonance studies have determined the macroscopic diffusion activation energy of $\beta$-$Li_3PS_4$ to be 0.40 eV[40]. Similar values have also been obtained in DFT and MD simulations as shown in the table. The ionic conductivity and activation energy of $\beta$-$Li_3PS_4$, glassy $Li_3PS_4$, and glass-ceramic $Li_3PS_4$ electrolytes obtained from our MLIP simulations are consistent with DFT calculated values and comparable to the experimental values. Regarding ionic conductivity, a previous study has shown that discrepancies in conductivity between simulations and experiments are due to grain boundaries[41]. Furthermore, although DFT calculations with different basis yield varying lattice parameters, these differences have little impact on the calculated diffusion coefficients[42]. Additionally, we provide a comparative analysis with experimental scattering data from Ref. 43. Supplementary Figure S1 shows the comparison between MD simulation and neutron and X-ray scattering data regarding the structure factor $S(Q)$ for glassy $Li_3PS_4$. The $R_X$ factors in Supplementary Fig. S1, as introduced by Wright[44], are calculated, confirming the agreement between the simulated and experimental scattering results, demonstrating a good agreement for the present MLIP.

### Structure descriptors for order and disorder

Figure 1b presents the contrasting spatial distributions of $Li^+$ and $PS_4^{3-}$ units within both crystalline $\beta$-$Li_3PS_4$ (top) and glassy $Li_3PS_4$ (bottom). In the $\beta$-$Li_3PS_4$ crystal, the $PS_4^{3-}$ groups are arranged in an orderly zigzag pattern. Adjacent $PS_4^{3-}$ groups construct tetrahedral lithium sites (termed Li1) and octahedral lithium sites (termed Li2)[45], as shown in the bottom panel of Supplementary Figs. S2a, b, which also include interstitial tetrahedral sites (termed Li3). These sites are mainly distributed in the *ac*-direction plane. There are two types of interstitials, termed i2 and i3 (distorted octahedral site)[46]. Lithium diffusion in $\beta$-$Li_3PS_4$ obtained from experiments and simulations is two-dimensional, i.e., in the *ac* plane, specifically the Li2-Li1-Li2 path. Studies also suggest that, due to the series of Li3 sites between different planes, lithium ions can deviate from the path in the *ac* plane, that is, along the *b* direction via the Li2-Li3-Li2 path[46]. Additionally, interstitial diffusion also occurs in the *ac* and *b* direction, such as Li-i2-Li1 diffusion path through interstitials between sites[46]. In contrast, the $PS_4^{3-}$ groups in glassy $Li_3PS_4$ electrolytes, which feature isotropic ionic conduction[19], exhibit a disordered arrangement (as illustrated in the upper panel of Supplementary Fig. S2c), making it difficult to define regular coordination sites and symmetrical migration paths[7]. Unlike in crystalline phase, the potential energy distribution of lithium-ions in glassy $Li_3PS_4$ is irregular, with ions exploring larger spaces by overcoming higher energy barriers[20]. The Gaussian density distribution projections of P and S atoms in the 2D plane (Supplementary Fig. S3) clearly illustrate the ordered arrangement of $PS_4^{3-}$ units in the crystal phase and their disorder in the glass phase.

Figure 1c shows snapshots of the glass-ceramic $Li_3PS_4$ (see *Methods* section for details on the construction of the glass-ceramic model using MD simulations), and a visual representation of the amorphization $F(Z)$ distribution along the *y*-axis of simulation cell. $F(Z)$ is

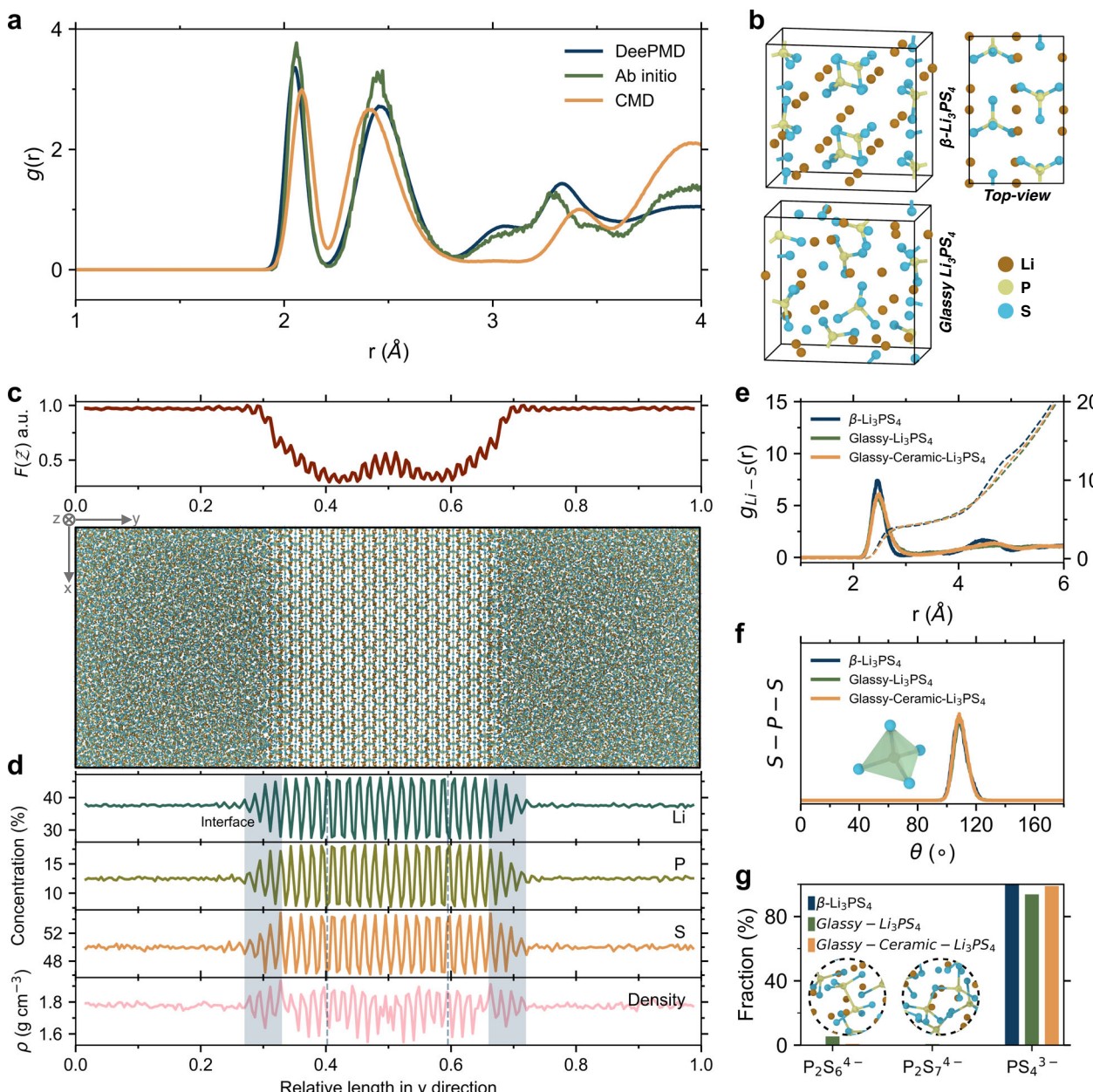

**Fig. 1 | Structural fingerprints in the transition from ordered to disordered electrolytes. a** Pair distribution function $g(r)$ of β-Li$_3$PS$_4$ electrolytes simulated by molecular dynamics simulations using machine learning interatomic potential (MLIP), DFT-based ab initio molecular dynamics (AIMD) simulations, and classical molecular dynamics (CMD) simulations[33]. **b** Atomic snapshots of β-Li$_3$PS$_4$ (top) and glassy Li$_3$PS$_4$ (bottom) electrolyte configurations. **c** Atomic snapshot of the glass-ceramic Li$_3$PS$_4$, depicting the amorphization distribution (Eq. 6) of the constructed glass-ceramic along the $y$-axis at the top. **d** Element concentration and density profiles for the glass-ceramic Li$_3$PS$_4$ from panel **c**. The gray rectangle and dashed line highlight the internal interface between the ordered and disordered phases and the exemplified crystalline plane. **e** Radial distribution function (RDF) of Li-S pairs and integrated RDFs for glassy-, β-, and glass-ceramic Li$_3$PS$_4$. The RDFs are shown as solid lines, while the integrated RDFs are represented by dashed lines. **f** Angular distribution function of S-P-S. **g** Fractions of thiophosphate anions in glassy-, β-, and glass-ceramic Li$_3$PS$_4$ system. Source data are provided as a Source Data file.

calculated from the Gaussian density of the atomic arrangement through the Fourier transform method (Eq. 6), as explained in the *Methods* section and Ref. 30. The glass phase has a $F(Z)$ value of approximately 1, with a transition of the degree of disorder occurring at the interface between the glass and crystal phases within the $F(Z)$ profile, that is, the transition zone where the curve exhibits a double-minima. Distinguished from the crystalline and glass phases, the clear definition of the interfacial region is derived from the numerical derivative $F(Z)'$ of the calculated $F(Z)$, as shown in Supplementary Fig. S4a, with the width of the interfaces quantified accordingly. This width is self-limiting, indicating that it does not depend on the

crystalline and glass phase content in the model, as verified in Supplementary Fig. S4b. The thermodynamically self-limited width of interfaces has also been observed in LATP electrolytes (Li$_{1+x}$Al$_x$Ti$_{2-x}$(PO$_4$)$_3$, a natrium superionic conductor-type solid electrolytes)[30]. The corresponding profiles of element concentration and density are depicted in Fig. 1d for the glass-ceramic Li$_3$PS$_4$ electrolyte. We compute averages along the $y$-axis using specific bin values derived from simulated boxes. The resulting profiles reveal the homogeneity of the glassy phase, whereas elements in the crystalline phase display a periodic distribution. Notably, the gray rectangle and dashed line highlight the internal interface, where the ordered and

disordered phases coexist, and an exemplified crystalline plane, respectively.

The glassy, crystalline, and glass-ceramic $Li_3PS_4$ are three solid-state electrolytes with distinctly different structures, i.e., varying degree of disorder. Their short-range order (SRO) and medium-range order (MRO) structures are characterized by the radial distribution function (RDF) $g(r)$ and neutron structure factor $S_N(Q)$, respectively. The RDF is defined as the probability density of finding another particle at a distance $r$, which can capture the local coordination of particles, structural periodicity, etc. RDFs of all the atom pairs are shown in Supplementary Fig. S5, showcasing a comparable distribution of P-S distances in the first peaks of both the disordered and ordered $Li_3PS_4$. The first peak is located at approximately 2.07 Å, corresponding to the length of P-S bonds[47,48]. We also discern a discernible periodicity in the distribution of Li-Li distances within the β-$Li_3PS_4$ sample. The RDFs of Li-S pairs presented in Fig. 1e provide the local coordination environment within the $Li_3PS_4$ systems, as quantified by the integrated RDFs. Despite the structural transition from order to disorder, the distance distributions of Li and S remain largely similar, with a coordination number of approximately 4 for Li. In β-$Li_3PS_4$, Li is situated within both tetrahedral ($LiS_4$) and octahedral lithium sites ($LiS_6$) among the S atoms. Conversely, in the disordered structure, the Li atoms in the four-coordinated configuration occupy a larger free volume (see Supplementary Fig. S6). The atomic volume of lithium is here characterized by the Voronoi tessellation method, which considers the region of real space closer to that of central particle[49–51].

The angular distribution function of S-P-S is shown in Fig. 1f, revealing the tetrahedral $PS_4^{3-}$ motifs formed between sulfur and phosphorous atoms. In glassy and glass-ceramic $Li_3PS_4$ electrolytes, the emergence of the P-P peak around 2.3 Å (Supplementary Fig. S5) is attributed to the formation of thermodynamically more stable $P_2S_6^{4-}$ motifs during the melt-quenching process of the disordered structure[52,53]. We quantify the fraction of these units in various systems, and as depicted in Fig. 1g, both $P_2S_6^{4-}$ and $P_2S_7^{4-}$ motifs are detected in glassy and glass-ceramic $Li_3PS_4$. For $Li_2S$-$P_2S_5$ glass electrolytes with different compositions, we also analyze how the number of different structural motifs varies with composition and pressure, as shown in Supplementary Fig. S7. This confirms that our MLIP is able to capture the reactions between $PS_4^{3-}$ units.

Since the scattering vector $Q$ is the inverse distance in real space, the low $Q$-values region of the $S(Q)$ encompasses structural information from SRO to MRO. Generally, different peak positions in the low $Q$-values region of $S(Q)$ correspond to ordering at different length scales. The first peak represents the arrangement of motifs in the medium range, the second peak reflects the size of local network-forming motifs, and the third peak provides information about nearest-neighbor interactions[54]. The first peak in the low $Q$-values region of $S(Q)$ is also known as the first sharp diffraction peak (FSDP), whose position is correlated with the size of the motif cluster with MRO through Ehrenfest's formula[55], and intensity reflects the degree of structural order[54]. As shown in Supplementary Fig. S8, a pronounced FSDP (higher intensity) of neutron $S_N(Q)$ is observed in β-$Li_3PS_4$, indicating a higher degree of structural order. Additionally, β-$Li_3PS_4$ exhibits FSDP at lower $Q$-values position compared to that in glassy $Li_3PS_4$, signifying that its structural order extends over a longer length scale as expected.

## Dynamics of hopping ions

The impact of disorder on lithium-ion mobility and transport properties is investigated by analyzing the dynamic diffusion behavior of hopping ions within the $Li_3PS_4$ systems. We first evaluate the time-averaged mean-squared displacement (MSD, $\langle r^2(t) \rangle$) of lithium ions (Fig. 2a). The MSD quantifies the average distance that the ions travel over time. The β-$Li_3PS_4$ electrolytes demonstrate the highest MSD at 900 K (Fig. 2a) and 1000 K (Supplementary Fig. S9a–c) for the same

time lag. Conversely, the glass-ceramic $Li_3PS_4$, comprising both glassy and crystalline phases, exhibits a marginally reduced MSD compared to the glassy $Li_3PS_4$. All $Li_3PS_4$ electrolytes feature diffusion dynamics at 1000 K, eventually reaching Fick's limit ($t^1$), as indicated by the exponent of MSD in Supplementary Fig. S9d, e (β- $Li_3PS_4$ reaches $t^1$ at 900 K as shown in Supplementary Fig. S9d).

As the temperature decreases, the cumulative displacement difference of the hopping lithium ions between β-$Li_3PS_4$ and both the glassy $Li_3PS_4$ and glass-ceramic $Li_3PS_4$ diminishes gradually. Specifically, the $Li^+$ MSD for β-$Li_3PS_4$ decreases with decreasing temperature, ultimately becoming smaller than that of both glassy and glass-ceramic $Li_3PS_4$ at 700 K (Supplementary Figs. S9a–c). Interestingly, the disordered structure exhibits the smaller decrease in MSD with decreasing temperature compared to the more ordered structure. This is due to the higher activation energy of diffusion in the crystal compared to that in glass and glass-ceramics (as further discussed in the section below). This means that the migration rate of lithium ions decreases more rapidly as the temperature is lowered in the crystal. Owing to the disordered structure, lithium-ion migration in glass and glass-ceramic does not have preferred pathways, resulting in isotropic transport. The diverse energy landscape[20] provides more potential ion migration paths, allowing for a higher number of mobile ions even as the temperature decreases. In contrast, lithium migration in crystals follows preferred pathways[46]. As the temperature decreases, the ionic mobility weakens, and lattice vibrations diminish, causing the migration channels to become more fixed, which significantly reduces the number of mobile ions.

Atoms may exhibit a displacement distribution that deviates from a Gaussian distribution, a phenomenon commonly referred to as dynamic heterogeneity[56]. This illustrates the situation where atoms within one domain exhibit notably faster movement compared to those in neighboring domains, typically separated by a few nanometers. The non-Gaussian parameter (NGP, $\alpha_2(t)$), or the fourth cumulant of displacement, has proven to be a measure of diffusion coefficient fluctuations and dynamic heterogeneity[57]. It characterizes the degree of deviation from the Gaussian behavior in particle diffusion. Specifically, the NGP calculated by Eq. 14 evolves over time, with the time point at which the NGP peak appears corresponding to when the system exhibits the maximum deviation from Gaussian behavior. The intensity of the NGP peak is a measure of the extent of this deviation, i.e., a higher peak intensity indicates a higher level of dynamic heterogeneity. For instance, some atoms move very quickly (a behavior known as "hopping"), while others move relatively slowly. It is important to note that the room ionic conductivity is associated with the static or structural heterogeneity, which is a frozen-in dynamic heterogeneity from high temperature. The position of the NGP peak (i.e., the time or displacement scale at which the peak occurs) is related to the timescale of non-Gaussian behavior in the system. For time-dependent NGP, the peak position is commonly used to characterize the timescales of different dynamic processes in the system, such as the glass transition, the relaxation of microstructures, and the fluctuations in the diffusion coefficient[57–59]. In this case, the displacement distribution of the system exhibits a great deviation from a Gaussian distribution.

For a more comprehensive understanding of the temporal dynamical events involving lithium-ions, we therefore employ the NGP descriptor to compare the dynamic heterogeneity within the different $Li_3PS_4$ systems. As shown in Fig. 2b, the motion of lithium ions exhibits non-Gaussian properties, indicating that it is not entirely random. Red squares denote the NGP peak times, symbolized as $\tau_{ngp}$, signifying the occurrence of non-Gaussian dynamics. An increase in the degree of disorder leads to a delay in the emergence of the NGP peaks and an increase in their peak heights. As illustrated in Supplementary Fig. S10, both the NGP peak height and time increase as the temperature decreases. We focus on the short-term

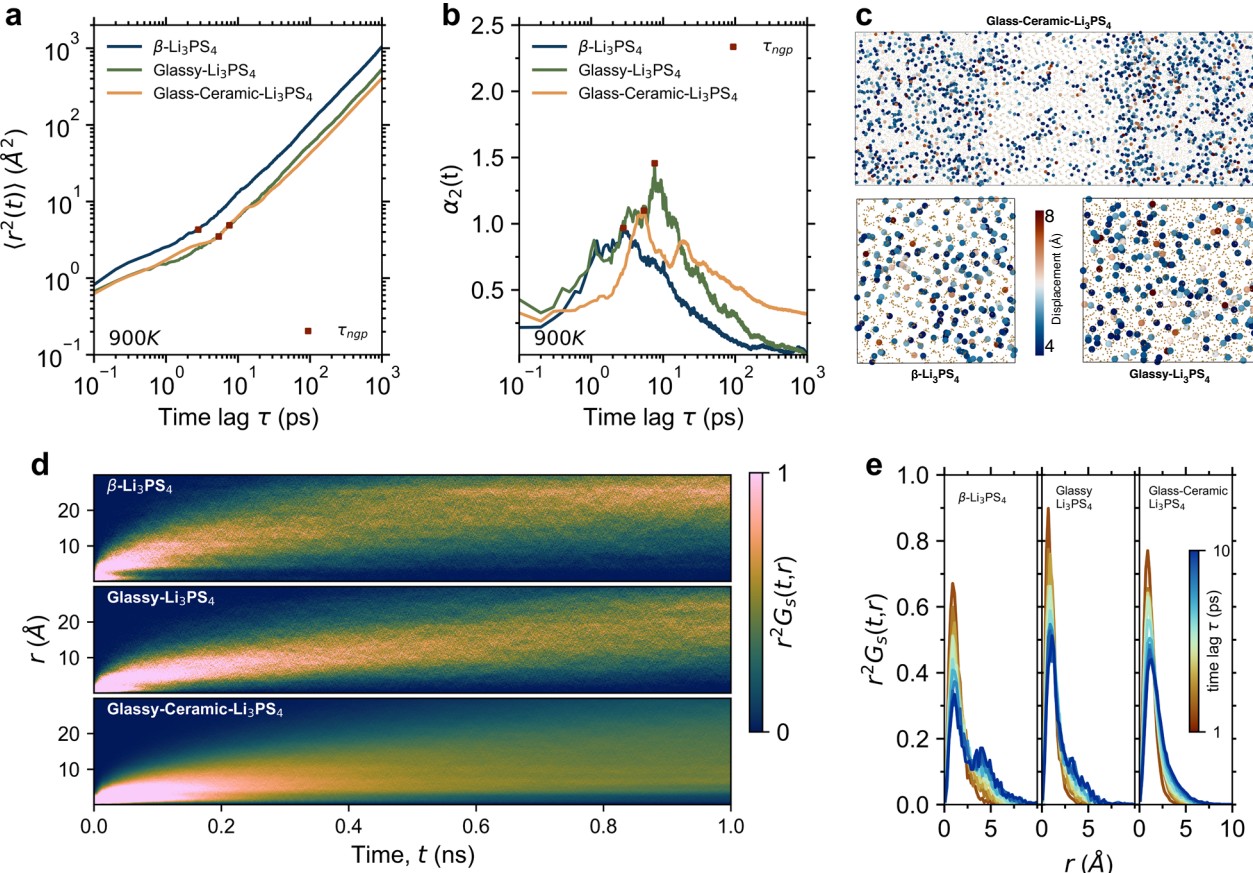

**Fig. 2 | Disorder-driven fast lithium diffusion. a** Time-averaged mean-squared displacement (MSD, $\langle r^2(t) \rangle$), and **b** non-Gaussian parameter (NGP, $\alpha_2(t)$) of lithium ions for β-$Li_3PS_4$, glassy $Li_3PS_4$, and glass-ceramic $Li_3PS_4$ systems, as functions of the time lag, $\tau$. The red squares represent the NGP peak times, $\tau_{ngp}$. **c** Short-term time-averaged displacement of lithium ions for 10 ps. The displacement magnitudes exceeding 4 Å are highlighted with larger markers for better representation. **d** Self-part van Hove correlation function for β-$Li_3PS_4$, glassy $Li_3PS_4$, and glass-ceramic $Li_3PS_4$ systems at 900 K. **e** Self-part van Hove correlation function for β-$Li_3PS_4$, glassy $Li_3PS_4$, and glass-ceramic $Li_3PS_4$ systems for fixed time between 1 and 10 ps at 900 K. Source data are provided as a Source Data file.

(10 ps) variations in lithium-ion displacement by enhancing the visualization of significant motions, as shown in Fig. 2c, where larger atoms are used to emphasize displacement magnitudes exceeding 4 Å. In the disordered glassy phase, there is an excess of lithium ions rapidly deviating from their initial vibrational positions, covering longer displacements compared to those in the crystalline phase.

To further explore the correlation between particle mobility for short-to-long term behavior, Fig. 2d, e showcase the self-part van Hove correlation function ($G_s$) of Li-Li for the different $Li_3PS_4$ system. $G_s(r,t)$ characterizes the Li-Li pair distance $r$ at time $t$, and the quantity $r^2G_s(r,t)$ describes the probability distribution of particle displacements[39]. As seen from Fig. 2d, β-$Li_3PS_4$ has higher probability distributions for long-distance displacements (>10 Å) with increasing time. This can be attributed to the extended displacements of hopping ions along specific transport pathways within the crystalline structure[46]. In contrast, glassy $Li_3PS_4$ performs less favorably due to the convoluted migration paths of ions within its disordered structure. When examining the probability distribution over a fixed time interval between 1 and 10 ps (Fig. 2e), the glass-ceramic $Li_3PS_4$ displays a single, wide peak distribution, centered at approximately 1 Å. This distribution is predominantly influenced by the equilibrium vibrations and occupation of nearest-neighbor sites[39]. Notably, within a time interval of only 10 ps, β-$Li_3PS_4$ undergoes a transition from a single peak to a double peak, indicating that lithium ions depart from their initial equilibrium positions to initiate migration. A subtle shift of the $r^2G_s(r,t)$ distribution is also observed in the glassy $Li_3PS_4$.

## Ionic conductivity

Macroscopic ionic conduction is the result of the collective ion migration dynamics within a system[19], Such dynamics depends on the structure of that system. In this context, here we illustrate the impact of the transition from ordered to disordered structure on the mobility of lithium ions and, consequently, on the derived ionic conductivity. We begin with the diffusion coefficient $D$, which can be calculated as the slope of the MSD-time curve. In Fig. 3a, the temperature dependence of the diffusion coefficient is illustrated, following an Arrhenius-type behavior at high temperature, with the degree of disorder influencing the temperature dependence. This is seen from the lower diffusion activation energy ($E_a$, as determined from the slope of the plot) in glassy and glass-ceramic $Li_3PS_4$ electrolytes as compared to that in β-$Li_3PS_4$. Below the glass transition temperature ($T_g$), glassy $Li_3PS_4$ gradually deviates from the Arrhenius behavior. This sub-$T_g$ non-Arrhenius behavior originates from the frozen structure of the glassy electrolytes and has been widely confirmed[42,60,61].

It is important to note that as the temperature decreases, the movement of lithium ions gradually shifts away from the diffusive region, exhibiting sub-diffusion, i.e., deviation from Brownian motion, as shown in Supplementary Fig. S11a. Therefore, due to the fact that the movement of particles under sub-diffusion is hindered by certain mechanisms (such as energy barriers, viscosity, etc.), the MSD ($\langle r^2(t) \rangle$) increases sub-linearly with time $t$ (i.e., $\langle r^2(t) \rangle \propto t^\alpha$, where $\alpha < 1$). As a result, the estimation of the diffusion coefficient is inaccurate. The extrapolated room temperature conductivity from high temperature

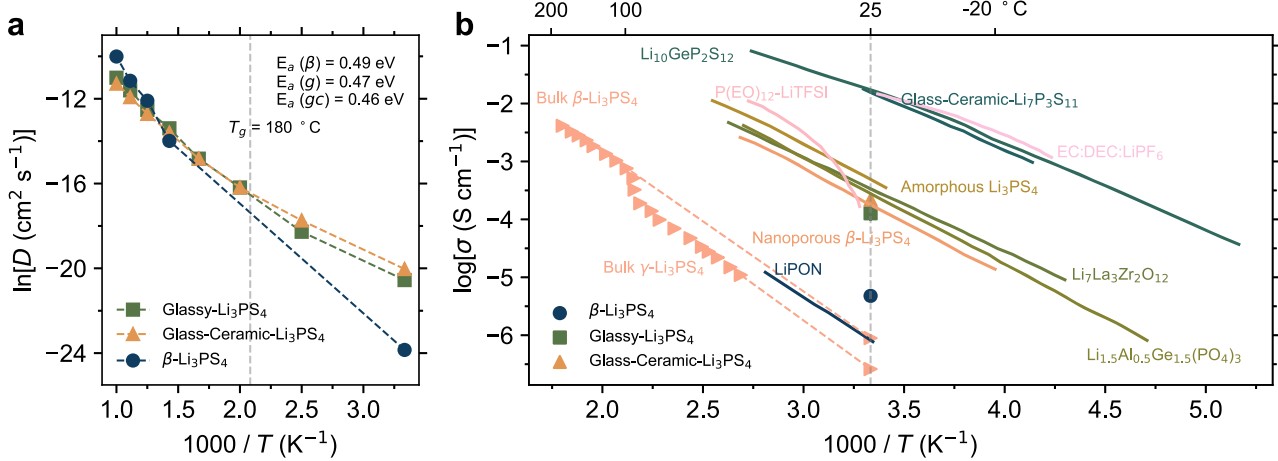

**Fig. 3 | Temperature dependent ionic conductivity. a** Temperature dependence of the lithium diffusion coefficient. **b** Comparison of room temperature ionic conductivity in $Li_3PS_4$ systems with varying degree of disorder, as well as results for other lithium solid electrolytes, organic liquid electrolytes, and polymer electrolytes[17,38,65,66,94–99]. Source data are provided as a Source Data file.

data is generally higher compared to simulations done at room temperature, as shown in Supplementary Fig. S11b.

As illustrated in Fig. 3b, the glassy $Li_3PS_4$ electrolytes exhibit notably high room-temperature ionic conductivity, i.e., higher than that of $\beta$-$Li_3PS_4$ electrolytes. Interestingly, glass-ceramic $Li_3PS_4$ electrolytes, which is characterized partially ordered structures, demonstrate $E_a$ and ionic conductivities comparable to, or even slightly superior to, those of purely glassy $Li_3PS_4$. The enhancement of ionic conductivity was also found in previous experimental work and calculations for both $Li_3PS_4$ system[16,62,63] and the $Li_2S$-$P_2S_5$ and $Li_7P_3S_{11}$ systems[15,17,23,26,64]. This enhancing effect consistently occurs in glass-ceramic $Li_3PS_4$ samples (see Supplementary Fig. S12). Specifically, the room-temperature ionic conductivities of glass-ceramic $Li_3PS_4$ electrolytes, varying in crystal contents, exceed that of glassy $Li_3PS_4$ electrolyte. Here, the crystallinity in Supplementary Fig. S12 is derived from the volume fraction of crystal in glass-ceramics. Although glass composition and relaxation effects are present in glass-ceramics, the influence of these effects is reduced due to the presence of crystalline phases, which help stabilize the material. Furthermore, it is interesting to note that the ionic conductivity of glass-ceramics $Li_3PS_4$ does not lie between those of glassy and crystalline $\beta$-$Li_3PS_4$ electrolytes. This arises from the interplay of glass-phase-induced and interfacial-phase-induced (disordered structures) mechanisms in glass-ceramics, as discussed in detail below. Figure 3b compares the room temperature ionic conductivity of the present $Li_3PS_4$ systems, which is calculated by means of the *Nernst-Einstein equation* (as described in the *Methods* section), with that of other types of ordered and disordered electrolytes. The simulated ionic conductivities of $Li_3PS_4$ with varying degrees of disorder align closely with previously reported experimental values[38,65,66].

**Disorder-induced ion transport enhancement in glass-ceramic structures**

We now focus on ion hopping dynamics in the glass-ceramic $Li_3PS_4$ electrolyte, which consists of both the glassy and crystalline phases, as well as the complex interface region. Figure 1 illustrates the interface involving a disordered transition region of $F(Z)$ and density variation divisions. These three phases exhibit varying degree of disorder, which can be quantified using the amorphization function $F(Z)$, and in simple terms, the degree of disorder follows the sequence: glass > interface > crystalline. We first compare the MSD results ranging from 300 K to 900 K (in 200 K intervals) of the three phases in the glass-ceramic $Li_3PS_4$. As shown in Supplementary Fig. S13 and Fig. 4a, we find that

both the glass phase and the interface outperform the crystalline phase significantly. The temperature sensitivity of the MSD in the crystal phase reveals that as temperature decreases, the MSD significantly drops as also observed for $\beta$-$Li_3PS_4$. The heterogeneous dynamics of different phases within glass-ceramic is characterized by NGP as presented in Fig. 4b. Combining with the NGP results of bulk $\beta$-$Li_3PS_4$, glassy $Li_3PS_4$, and glass-ceramic $Li_3PS_4$ presented in Supplementary Fig. S14, we infer that an increase in disorder degree is associated with that in the NGP peak intensity, leading to a more pronounced deviation of ion dynamics from a Gaussian distribution. It is evident from $G_s$ of Li-Li that in the glassy (Fig. 4c) and interfacial (Supplementary Fig. S15a) regions of the glass-ceramic $Li_3PS_4$, there is a significant long-range displacement of hopping ions over time at 900 K. In contrast, the crystalline phase of the glass-ceramic does not exhibit the probability distribution of $\beta$-$Li_3PS_4$. Instead, it demonstrates a high probability of displacements within the range of 3–10 Å throughout the entire simulation period (highlighted as the bright regions in the bottom panel of Fig. 4c).

The above results indicate that, the mobility of lithium ions within the disordered structure of the glass-ceramic (including the glassy and interfacial phases) is significantly superior to that in the crystalline phase. We find that this mobility discrepancy facilitates lithium-ion penetration, and the disordered structure enhances the dynamics driving the diffusion towards the crystalline phase, thereby increasing the content of mobile ions in the crystalline phase. The ionic conductivity in glass-ceramic $Li_3PS_4$ electrolyte is more than two orders of magnitude greater than that of $\beta$-$Li_3PS_4$. In the following, we further analyze the lithium-ion exchange among the various phases within the glass-ceramic $Li_3PS_4$. Figure 4d shows the time-concentration profiles of lithium ions in the various phases within the glass-ceramic $Li_3PS_4$, confirming that there is no significant enrichment of lithium ions in any phase during the diffusion process.

Figure 4e provides a visual representation of the trajectory of a single lithium-ion, highlighting the migration pathways of the lithium ions within the different phases of glass-ceramic $Li_3PS_4$. Specifically, in the disordered structure, ions navigate through the disordered potential-energy landscape[20], resulting in their meandering trajectories. Conversely, in the more ordered crystalline phase, distinct site-to-site hopping trajectories are apparent. In our analysis of the MSD within $\beta$-$Li_3PS_4$ electrolytes, we observe that the component along the $z$-direction is superior to that along the $y$-direction and significantly outperforms the $x$-direction (Supplementary Fig. S16). This phenomenon is consistent with the results reported in Ref. 46, where the

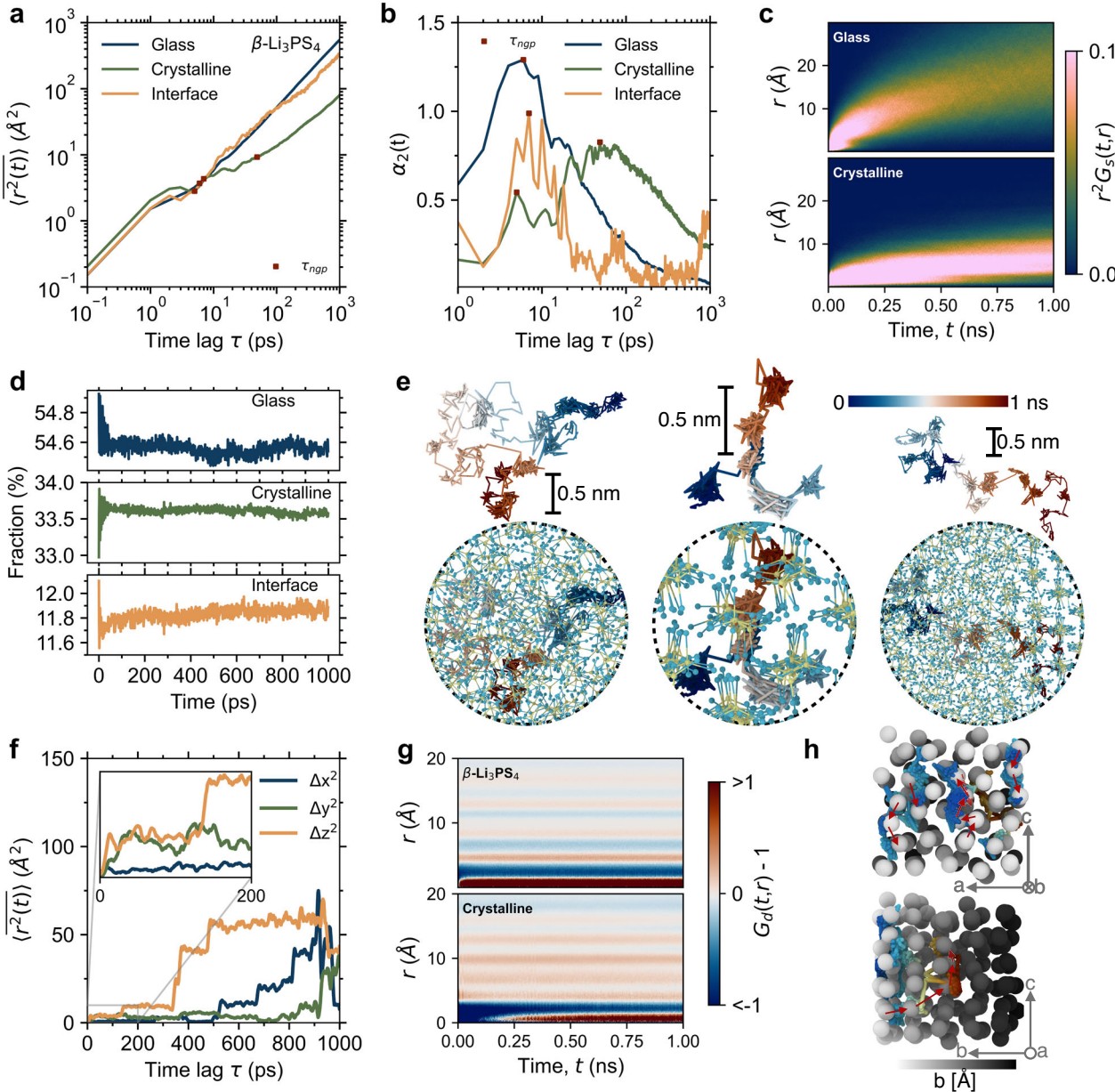

**Fig. 4 | Diffusion dynamics from order to disorder. a** Time-averaged mean-squared displacement (MSD) and **b** non-Gaussian parameter (NGP) of lithium ions for glass-ceramic $Li_3PS_4$ system. **c** Self-part van Hove correlation function for glass-ceramic $Li_3PS_4$ system. **d** Time profiles of lithium-ion fraction within each phase of the glass-ceramic $Li_3PS_4$ system. **e** Atomic snapshots of lithium ions migration trajectories within glass (left), crystalline (center), and interfacial (right) phases of the glass-ceramic $Li_3PS_4$ system over a timespan of 1 ns. **f** Short-term MSD of crystalline phase in glass-ceramic $Li_3PS_4$. **g** Distinct-part van Hove correlation function for $\beta$-$Li_3PS_4$ (top) and crystalline phase in glass-ceramic $Li_3PS_4$ (bottom). **h** Schematic diagram of lithium-ion hopping paths from different perspectives in the crystalline phase, where the lithium-ions and the selected lithium hopping trajectories are colored according to their depth in the $b$-direction. Source data are provided as a Source Data file.

lithium migration path in $\beta$-phase $Li_3PS_4$ crystals is found to primarily occur in the $ac$ plane, with additional migration paths existing in the $b$ direction as discussed above. In glass-ceramic $Li_3PS_4$ electrolytes, the fraction of lithium ions in each phase changes significantly on a short time scale, indicating ion penetration, and then stabilizes, reaching dynamic equilibrium (Fig. 4d). The fact that the MSD component in the $y$-direction for the glassy and interfacial phases is significantly higher than the other components further supports this migration preference (Supplementary Fig. S17). The dynamics within the disordered structure induce the ability for lithium ions in the crystalline phase of the glass-ceramic $Li_3PS_4$ to diffuse along the $y$-direction in the short term (see inset of Fig. 4f), ultimately leading to lithium-ion exchange

between different phases. Over longer timescales, lithium-ion migration in the crystalline phase is promoted along the $x$ and $z$ directions (Fig. 4f), i.e., the $ac$ plane.

Based on these results, we propose that the high ionic conductivity induced by the disordered phase arises from its facilitation of lithium-ion exchange between the crystalline and disordered phases, as well as the enhanced cooperative migration of lithium ions within the crystalline phase. This cooperative migration is also evidenced by the distinct-part of the van Hove correlation function ($G_d(r,t)$), which characterizes the probability of finding all other lithium ions at a distance $r$ from lithium-ion $j$ over time $t$. As presented in Fig. 4g and Supplementary Fig. S18, whether at high temperature or room

temperature, the cooperative migration in the crystalline phase of the glass-ceramic $Li_3PS_4$ is enhanced compared to that in the β-$Li_3PS_4$ and glassy $Li_3PS_4$ electrolytes, further proving the disorder-driven effect. Figure 4h visualizes the cooperative migration of lithium ions in the crystalline phase of glass-ceramic $Li_3PS_4$ along the *ac* plane and the *b* direction. It is important to note that the lithium migration dynamics driven by disordered structures are less affected by crystal orientation. As shown in Supplementary Fig. S19, this driving force enhances ion exchange between different phases, accelerating lithium-ion migration in the crystalline phase of glass-ceramics, including mechanisms such as cooperative migration.

### Identifying ion conducting dynamics in both ordered and disordered structures

In the case of crystals, ionic conductivity are related to the charge, concentration, and mobility of conducting ions[20]. The mechanism behind ion conduction can be explained through the hopping theory of conducting ions. In disordered structures, the lack of traditional coordination site and symmetric remote pathways[7], and the necessity for a percolating pathway of sites to minimize coordination changes[67,68], imply that the ion conduction is localized. That is, the ions are hopping between different sites, which possess different local environment. In both cases, the local coordination environment strongly impacts the ionic conduction. Given the complexity of local structures in disordered systems, it is impossible to identify all potential descriptors. Machine learning greatly accelerates and simplifies this process. Recently, the development of machine learning

algorithms has enabled the prediction of atomic dynamic properties based solely on the local structure and rearrangement capability of atoms from static structures[36,69].

Here, we employ a classification-based machine learning approach, referred to as 'softness'[35-37], to establish a connection between the dynamics of hopping ions and the degree of structural order. Our previous work[70,71] has demonstrated the effectiveness of the softness method in capturing the local structural features of glassy electrolytes and establishing correlations between structure and the dynamics of conducting ions. To calculate softness, we first analyze the static structures and corresponding rearrangements of each lithium ion at 300 K. Subsequently, we employ logistic regression to establish a hyperplane for distinguishing 'mobile' from 'immobile' lithium ions, thereby determining their mobility characteristics, with softness ($S$) being defined as the distance to the feature space hyperplane. The mobility is analyzed using the non-affine square displacement ($D_{min}^2$) for each lithium ion, with the sum of rearrangements for each lithium ion ($D_{cum}$) being used to quantify the extent of atomic rearrangements. For more information, we refer to the *Methods* section.

We plot the distribution of lithium softness $S$ values for β-$Li_3PS_4$, glassy $Li_3PS_4$ and glass-ceramic $Li_3PS_4$ in Fig. 5a. Positive values of $S$ correspond to mobility, whereas negative values signify immobility. The increase in disorder degree indeed shifts the softness distribution towards greater mobility, consistent with the observations reported above. The glass-ceramic $Li_3PS_4$ exhibits a bimodal distribution of $S$, with each peak corresponding to the softness distribution of immobile crystalline and mobile glassy phases, respectively, as illustrated in

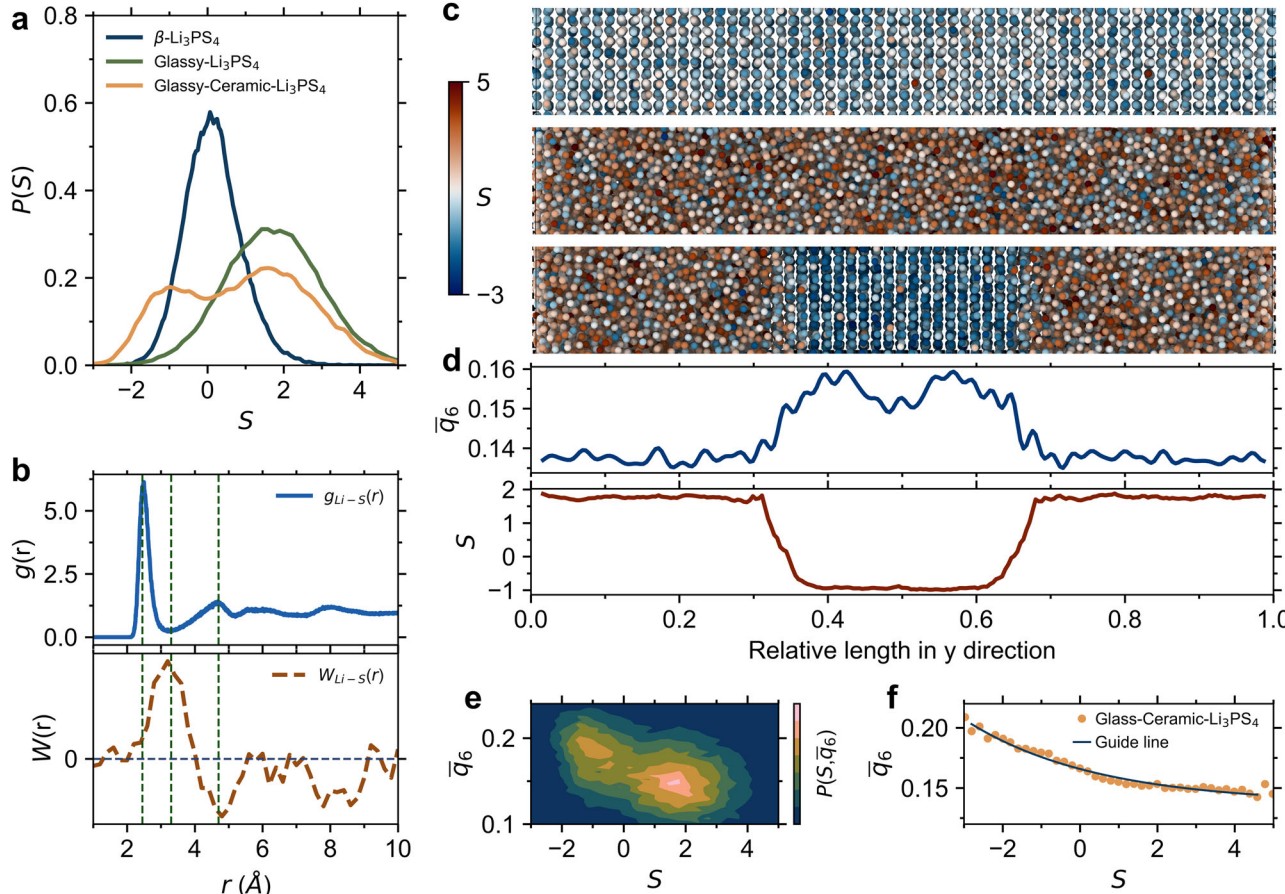

**Fig. 5 | Machine-learning classified softness parameter. a** Distribution of lithium softness $S$ in β-, glassy-, and glass-ceramic $Li_3PS_4$ systems at 300 K. **B** Radial distribution function (top panel) and weight function (bottom panel) of the Li-S pair for glass-ceramic $Li_3PS_4$ system. **c** Atomic snapshots of softness distribution in β-, glassy-, and glass-ceramic $Li_3PS_4$ systems. **d** Profiles of the averaged Steinhardt order parameter $\bar{q}_6$ and lithium softness $S$ along the *y*-direction in the glass-ceramic $Li_3PS_4$ configuration. **e** Density distribution of lithium softness $S$ and the averaged Steinhardt order parameter $\bar{q}_6$. **f** Correlation between the lithium softness $S$ value and $\bar{q}_6$. Source data are provided as a Source Data file.

Supplementary Fig. S20. This result is in line with the lower lithium-ion diffusion capability observed in crystalline phases compared to the glassy phases within glass-ceramic $Li_3PS_4$.

The classification of lithium-ion mobility relies solely on the radial structure functions of Li-Li, Li-P, and Li-S pairs, achieving an accuracy of approximately 80%. Among these pairs, the $g(r)$ of Li-S for glass-ceramic $Li_3PS_4$ is the key correlation function to reflect the local structural environment of lithium ions, and hence dominates overall mobility. This aspect is illustrated in Fig. 5b, where $W(r)$ corresponds to the importance of each feature of $g(r)$. At the first peak of $g(r)$, the Li-S distance is at the equilibrium position, making it difficult for lithium ions to undergo significant rearrangement. As the distance increases, the Li-S separation starts to deviate from this equilibrium position. The increase in $W(r)$ indicates that the lithium ions become more likely to rearrange, reflecting a higher degree of 'softness'. In addition, we can observe positive values of $W(r)$ within the range of 6–8 Å, indicating that the MRO structure also influences the lithium-ion mobility. By coloring the lithium atoms based on their $S$ values, we can visualize the spatial softness distribution within the $Li_3PS_4$ structure (Fig. 5c), revealing a strong correlation between high $S$ values and structural disorder.

In the case of the glass-ceramic $Li_3PS_4$ electrolyte, we calculate the evolution of $S$ values within their structural landscape as the degree of disorder varies. To this end, we calculate a local bond-order parameter, known as the Steinhardt order parameter[72]. In detail, we employ the global average $\bar{q}_6$ parameter to characterize the structural order[73], which involves the first and the second shell. Figure 5d shows the calculate profile, demonstrating a strong correlation between $\bar{q}_6$ and $S$ values as the disorder-to-order transition occurs within the glass-ceramic $Li_3PS_4$ structure, with an abrupt transition at the interface. The $w_l$ order parameter, a variant Steinhardt order parameter, has also been used for comparison. We find that the same degree of $S$ values is observed during the transition between ordered and disordered states (Supplementary Fig. S21).

Figure 5e presents the density distribution of $\bar{q}_6$ and softness values, with density values color-coded according to the magnitude of softness. High-softness regions are highlighted as areas with lower $\bar{q}_6$ values, while ions with low $S$ exhibit relatively higher $\bar{q}_6$ values. Therefore, we conclude that there exists a strong correlation between the structural order and the $S$ parameter, as quantified using the curve in Fig. 5f. This negative correlation between the $S$ values and the local structural order corroborates the enhanced dynamical properties of hopping ions in disordered environments. This approach is versatile and applicable not only to glassy and crystalline $\beta$-$Li_3PS_4$ electrolytes, as demonstrated in Fig. 5, but also to $Li_3PS_4$ systems with varying crystal contents and different crystal orientations, as shown in Supplementary Fig. S22 (cell parameters of $\beta$-$Li_3PS_4$ are shown in Supplementary Fig. S23). This means that we can use this approach to predict the dynamic characteristics of the different phases solely based on their static structures, effectively distinguishing various phases with different degrees of softness.

## Discussion

We have highlighted the enhancing effect of structural disorder on the dynamics of hopping ions and the synergistic role of disordered glass and interface phases in promoting ion conductivity. This holds significant implications for the conduction mechanisms of ionic species in solid-state electrolytes, as disordered solid-state electrolytes with complex atomic structures are among the most promising ones. However, due to the lack of long-range order, analyzing their structure is challenging. In this study, we have trained a MLIP that balances high accuracy and efficiency and captures both ordered and disordered phases, allowing us to explore lithium-ion transport behavior on large scales and over long timescales. Combined with the machine learning-based classification method to uncover hidden structural information, this approach provides a robust template for studying ion transport behavior and mechanisms in other solid-state electrolytes, particularly in disordered glassy and partially disordered glass-ceramic electrolytes, such as LATP glass-ceramics[30]. Thus, linking dynamical characterics with local structural information and atomic rearrangements holds promise in unraveling ion transport mechanisms and discovering potential solid electrolytes for all-solid-state batteries.

Our work further indicates that disordered $Li_3PS_4$ solid-state electrolytes exhibit non-Arrhenius behavior at low temperatures and superior room-temperature conductivity compared to their crystalline counterparts. The dynamic heterogeneity of lithium ions plays a crucial role since the dynamic heterogeneity can be frozen-in upon cooling, leading to static or structural heterogeneity that impacts the room-temperature ionic conductivity. The partially disordered glass-ceramic demonstrates the highest room-temperature conductivity. From the atomic scale, we demonstrate that this superior ion conduction performance relative to the glass and crystalline electrolytes originates from the disorder-driven diffusion dynamics. This means that the dynamic interplay between disordered glass phases and disordered interfaces enhances ion exchange between the crystalline phase and other phases, with lithium-ion migration in the crystalline phase exhibiting enhanced cooperative diffusion.

## Methods

### General methods

We employed the Python packages NumPy[74], Pandas[75], and SciPy[76] for data processing and calculations, and utilized Matplotlib[77] and Ovito[78] for graph generation and the creation of renderable structural visualizations.

### Ab initio simulations

The initial datasets for training the MLIP were generated from the trajectories of ab initio molecular dynamics (AIMD) simulations. To increase the generalizability of the MLIP, series of systems containing Li, P, and S elements were covered in the datasets, including both the crystalline and disordered structures of elementary substances and compounds, such as Li, P, S, $Li_2P_2S_6$, $\beta$-$Li_3PS_4$, and $\gamma$-$Li_3PS_4$, as well as $x$$Li_2S$·$(100-x)$$P_2S_5$ ($x = 67, 70, 75$, and $80$) glasses. Detailed information on the included systems can be found in the Supplementary Table S2. The AIMD calculations were carried out at the DFT level[79] using the Quickstep module of the CP2K package[80] with the hybrid Gaussian and plane wave method (GPW)[81]. To ensure computational accuracy, the basis functions were mapped onto a multi-grid system with the default number of four different grids with a plane-wave cutoff for the electronic density of 500 Ry and a relative cutoff of 50 Ry. We performed the convergence test on the employed basis sets. The accuracy of the DFT calculations thus heavily relied on the grid size, which is defined by the planewave cutoff (cutoff) and the relative cutoff at which a Gaussian is mapped (Rel_cutoff). These two cutoff values should be high enough to achieve an accurate calculation. To this end, we systematically investigated the dependence of total energy of $\beta$-$Li_3PS_4$ on these two values. As shown in Supplementary Fig. S24, the plane-wave cutoff of 500 Ry and relative cutoff of 50 Ry ensures the accuracy of the DFT calculations.

The AIMD trajectories at 3000 K were obtained in the $NVT$ ensemble with a timestep of 0.5 fs for 2.5 ps. The temperature selection of 3000 K enabled the sampling of the melting process within the short time scale, which can be used for simulating both the crystal and glass structure afterwards. The temperature was controlled using the Nosé–Hoover thermostat[82]. The exchange-correlation energy was calculated using the Perdew-Burke-Ernzerhof (PBE) approximation[83], and the dispersion interactions were handled by utilizing the empirical dispersion correction (D3) from Grimme[84]. The pseudopotential GTH-PBE combined with the corresponding basis sets was employed to describe the valence electrons of Li (DZVP-MOLOPT-SR-GTH), P (TZVP-MOLOPT-GTH), and S (TZVP-MOLOPT-GTH), respectively[85].

## MLIP training and validation

The MLIP was developed using the DeepMD-kit[34] software combined with the active machine learning method implemented in the DP-GEN package[86]. The training process follows our previous work on zeolitic imidazolate framework glasses[87]. The training dataset consisted of two parts: (1) the initial training data from the trajectories of AIMD simulations on various Li, P, and S-containing systems at 3000 K for 2.5 ps, which allows the exploration of the energy landscape in an efficient manner; and (2) the expanded dataset realized by single energy calculation of different $x$Li$_2$S-(100-$x$)P$_2$S$_5$ ($x$ = 67, 70, 75) glasses using the active machine learning method implemented in the DP-GEN package[86]. The detailed information of the two datasets can be found in the Supplementary Tables S2 and S3, respectively. The network structure used for training the MLIP also consisted of two parts: (1) the embedding network with 3 layers of neurons (25, 50, 100); and (2) the fitting network with 3 layers of neurons (240, 240, 240). The local environment of the individual atoms was described using the descriptor containing both radial and angular information within a cutoff of 6.5 Å. The two training processes were adopted to increase the accuracy of the MLIP, i.e., the initial training and final training both included 6,000,000 steps of iterations. The energy, force, and virial terms were included in the loss function, which enabled the MLIP to work for both the structure and mechanical simulations. For the initial training, the learning rate dynamically changed from 1e−3 to 1e−9. The atomic interactions beyond 0.9 Å were treated using the ZBL repulsive interactions to avoid the collapse of atoms at high temperatures. The prefactors of the energy, force, and virial terms dynamically changed from 0.02 to 2, 1000 to 1, and 0.02 to 0.2, respectively. During the final training, the ZBL interaction was removed and the prefactors of all terms were set to 1 as the learning rate changed from 1e−5 to 1e−8. The performance of the MLIP model on different Li$_3$PS$_4$ phases ($\alpha$, $\beta$, and $\gamma$ phases) is shown in Supplementary Fig. S25, providing compelling evidence of its ability to accurately describe interatomic interactions. Our MLIP also contains the information regarding crystal-to-amorphous transition (Supplementary Fig. S26), which can predict the interfacial structure of $\beta$-Li$_3$PS$_4$.

## MD simulations

MD simulations were performed using the Large-scale Atomic/Molecular Massively Parallel Simulator (LAMMPS)[88]. The neural network potential trained using the DeepMD method, as described above, was employed to accurately describe the interatomic potentials. The temperature and pressure were controlled using Nosé-Hoover[82] thermostat/barostat methods. $\beta$-Li$_3$PS$_4$ with the *Pnma* space group was annealed from 500 K and relaxed for 1 ns at 300 K using the *NPT* ensemble. The preparation of glassy Li$_3$PS$_4$ was done using melt-quenching by initially raising the system to 1500 K, holding it for 100 ps, and subsequently cooling it down to 300 K at a rate of 2.5 K/ps before relaxation, all under the *NPT* ensemble. A time step of 0.5 fs was employed for precise and stable structural simulations throughout. The glass-ceramic Li$_3$PS$_4$ was made analogously to the glass, with a melt-quenching process under the *NVT* ensemble, during which the crystalline region was frozen (with all force components set to zero), followed by relaxation for glass and crystalline region under the *NPT* ensemble.

## Structural descriptors

The partial radial distribution functions $g_{ij}(r)$ define the probability of finding a particle $j$ at distance $r + \Delta r$ given that there is a particle $i$,

$$g_{ij}(r) = \frac{n_{ij}(r)}{4\pi r^2 \mathrm{d}r \rho_j} \tag{1}$$

where $n_{ij}$ is the number of $j$-type atoms found in a spherical shell of radius $r$ and thickness $\Delta r$, with the centra $i$-type atom. $\rho_j$ is the number density of $j$-type atoms. The running coordination n($r$) is obtained by integrating $g_{ij}(r)$ between $r_1$ and $r_2$ as,

$$n(\mathrm{r}) = \int_{r_1}^{r_2} 4\pi r^2 \rho_j g_{ij}(r) \mathrm{d}r \tag{2}$$

The theoretical partial structure factor $S_{ij}(Q)$ was calculated using the Faber Ziman formalism[89] as,

$$S_{ij}(Q) - 1 = \rho \int_0^\infty 4\pi r^2 \left(g_{ij}(r) - 1\right) \frac{\sin Qr}{Qr} \mathrm{d}r \tag{3}$$

where $Q$ is the scattering vector, and the X-ray scattering structure factor $S_x(Q)$ was calculated as,

$$S_X(Q) = \sum_i \sum_{j=i} c_i c_j f_i(Q) f_j(Q) \left(S_{ij}(Q) - 1\right) \tag{4}$$

Here, $c_i$ and $c_j$ are the atomic fractions, $f_i(Q)$ and $f_j(Q)$ refer to the $Q$-dependent X-ray scattering coefficients of type $I$ and $j$ atoms, respectively. By considering the coherent scattering length $I$ and $\bar{b}_j$ of atoms, the neutron scattering structure factor $S_N(Q)$ was calculated as,

$$S_N(Q) = \sum_i \sum_{j=i} c_i c_j \bar{b}_i \bar{b}_j \left(S_{ij}(Q) - 1\right) \tag{5}$$

The degree of disorder was quantified using the function $F(Z)$. We followed a procedure similar to that in Ref. [30], beginning with a 3D Gaussian density distribution of the atomic positions. After mapping these positions onto a 3D grid, we created density slabs along the $y$-axis of the simulation box with a width of $\Delta y$ and projected them into 2D. By summing the intensities, which were obtained through a 2D discrete Fourier Transform with the zero-frequency component shifted to the center, we calculated $F(Z)$, which was then normalized to its maximum value,

$$F(z) = \frac{\sum_{xz} I_{2D-FFT}(y)}{\sum_{xz} I_{max}} \tag{6}$$

To distinguish between ordered crystalline phases and disordered glassy phases, we introduced an average local bond order parameter, or the average Steinhardt order parameter. The local rotationally invariant $q_l$ or $w_l$ order parameter described by Steinhardt were implemented as follows. First, based on the spherical harmonic algorithm, a complex vector for a particle can be defined as $q_{lm}$ (Eq. 7), where $N_b$ is the number of nearest neighbors for particle $i$, and $Y_{lm}$ represents spherical harmonics,

$$q_{lm}(i) = \frac{1}{N_b} \sum_{j=1}^{N_b} Y_{lm}(\theta(\vec{r}_{ij}), \phi(\vec{r}_{ij})) \tag{7}$$

The Steinhardt order parameters[72] are defined as,

$$q_l(i) = \sqrt{\frac{4\pi}{2l+1} \sum_{m=-l}^{l} |q_{lm}(i)|^2} \tag{8}$$

The contributions of each neighbor are weighted based on Voronoi tessellation:

$$q'_{l_m}(i) = \frac{1}{\sum_{j=1}^{N_b} w_{ij}} \sum_{j=1}^{N_b} w_{ij} Y_{lm}(\theta(\vec{r}_{ij}), \phi(\vec{r}_{ij})) \tag{9}$$

Thus, all $q_{lm}$ calculations are replaced by the weighted $q'_{lm}$. Here, we performed a second averaging over the first shell of particles, implicitly incorporating information from the second shell[73], which was computed by replacing $q_{lm}(i)$ with $\bar{q}_{lm}(i)$. The average value of $q_{lm}(i)$ over all the $N_b$ neighbors $k$ of particle $i$, including particle $i$ itself, was calculated as,

$$\bar{q}_{lm}(i) = \frac{1}{N_b} \sum_{k=0}^{N_b} q_{lm}(k) \tag{10}$$

We also calculated the $w_l$ order parameter[72], which is defined as a weighted average over the $q_{lm}(i)$ values using Wigner 3-j symbols (related to Clebsch-Gordan coefficients). The resulting combination is rotationally invariant:

$$w_l(i) = \sum_{m_1 + m_2 + m_3 = 0} \begin{pmatrix} l & l & l \\ m_1 & m_2 & m_3 \end{pmatrix} q_{lm_1}(i) q_{lm_2}(i) q_{lm_3}(i) \tag{11}$$

The $w_l$ was normalized as:

$$w_l(i) = \frac{\sum_{m_1 + m_2 + m_3 = 0} \left( \begin{smallmatrix} l & l & l \\ m_1 & m_2 & m_3 \end{smallmatrix} \right) q_{lm_1}(i) q_{lm_2}(i) q_{lm_3}}{\left( \sum_{m=-l}^{l} |q_{lm}(i)|^2 \right)^{3/2}} \tag{12}$$

## Lithium transport dynamics

The mean-squared displacement (MSD) and non-Gaussian parameter were calculated from long-time lag ($t$) trajectory as,

$$\overline{\langle r^2(t) \rangle} = \Delta r^2(t) = \left\langle (r(t) - r(0))^2 \right\rangle \tag{13}$$

$$\alpha_2(t) = \frac{d \left\langle (r(t) - r(0))^4 \right\rangle}{(d+2) \left\langle (r(t) - r(0))^2 \right\rangle^2} - 1 \tag{14}$$

where the angular brackets denote an ensemble-averaged over the total conduction atoms, i.e., Li ions, and $d$ is the dimension of the simulation box, where $d = 3$ for all simulations. The self-part of the van Hove correlation function $G_s$ for Li-Li pair was calculated as,

$$G_s(r, t) = \frac{1}{4\pi r^2 N} \left\langle \sum_{i=1}^{N} \delta \left[ r - |r_i(t_0) - r_i(t + t_0)| \right] \right\rangle_{t_0} \tag{15}$$

where $G_s(r,t)$ characterizes the Li-Li pair distance $r$ at time $t$, and the quantity $r^2 G_s(r,t)$ describes the probability distribution of displacements. The distinct-part of van Hove correlation function $G_d(r,\text{t})$ characterized the real-space radial distribution function of distinct particles over time $t$ with respect to the initial reference particle. $G_d(r,t)$ is crucial for studying cooperative migration[90], and it was calculated as,

$$G_d(r, t) = \frac{1}{4\pi r^2 \rho N} \left\langle \sum_{i \neq j}^{N} \delta \left[ r - |r_i(t_0) - r_j(t + t_0)| \right] \right\rangle_{t_0} \tag{16}$$

The self-diffusion coefficient $D$ was estimated from the MSD and hence the activation energy $E_a$ was calculated by fitting an Arrhenius function as,

$$D = \frac{1}{2d} \lim_{t \to \infty} \frac{d\langle \text{MSD} \rangle}{dt} \tag{17}$$

$$D = D_0 \exp(-E_a / k_B T) \tag{18}$$

where $k_B$ is the Boltzmann constant, $T$ is the temperature, and $D_0$ is the self-diffusion coefficient at an infinite temperature. Finally, by means of Nernst-Einstein equation using the elementary charge $e$, the ionic conductivity $\sigma$ was calculated,

$$\sigma = \frac{N}{V} \frac{(Ze)^2}{k_B T} D \tag{19}$$

## Machine learning classification

We calculated the 'softness' metric[35–37] based on classification-based machine learning, following the procedure outlined in Refs. [70,71]. Unlike the original concept of softness, the computation of softness here relies on a logistic regression classifier instead of a support vector machine due to its higher classification accuracy and training efficiency[70,71,91]. Softness is defined as the distance to the feature space hyperplane, with radial order parameters chosen as the features for constructing the hyperplane. The hyperplane created by logistic regression can then be expressed as a function of each feature,

$$\sum_r W(r) G(i; r) - b = 0 \tag{20}$$

where the feature $G(i, r)$ represents the standardized radial order parameters, being a function of pairwise distance $r$. Here, $W(r)$ and $b$ are the weight coefficients and bias of the logistic regression model, respectively. The hyperplane is a linear combination of input features, allowing softness to be determined based on different features. In other words, the absolute value of $W(r)$ indicates the importance of the corresponding feature $G(i, r)$, with positive and negative signs signifying that an increase in the value of $G(i, r)$ will, respectively, increase or decrease the softness value.

The logistic regression model's output results were chosen based on the $D_{cum}$ metric for each lithium atom to analyze ion conduction behavior. The $D_{cum}$ is the sum of square-root of the incremental non-affine squared displacement $D_{min}^2$, which has been widely used to describe atomic rearrangement processes[35,92]. We optimized the intervals ($dr$) and cutoff radius ($R_{cutoff}$) of the radial order parameters based on their classification accuracy, as illustrated in Supplementary Fig. S27. By employing $dr$ of 0.2 Å and $R_{cutoff}$ of 10 Å, we established structural features that yielded prediction accuracy exceeding 0.78 for the test set. Regularization parameters $C$ and the threshold for $D_{cum}$ were also determined based on classification accuracy, as shown in Supplementary Fig. S28. Both the training and test sets achieved accuracies exceeding 80%. With these adjustments, we have constructed a framework capable of predicting the ionic conductivity dynamics based on the softness properties.

## Data availability

The templates to generate glass and glass-ceramic structures and to run simulations, and neural network potential file are available at https://github.com/OxideGlassGroupAAU/LiPS[93]. Source data are provided with this paper.

## Code availability

LAMMPS and DeePMD-kit are free and open-source codes available at https://lammps.sandia.gov and https://github.com/deepmodeling/deepmd-kit, respectively.

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

## Acknowledgements

This work was supported by grants from the European Union (ERC, NewGLASS, 101044664) and China Scholarship Council (202106880010). Views and opinions expressed are, however, those of the authors only and do not necessarily reflect those of the European Union or the European Research Council. Neither the European Union nor the granting authority can be held responsible for them. We also acknowledge the computational resources supported by EuroHPC Joint Undertaking with access to Vega at IZUM, Slovenia (EHPC-REG-2022R02-224), National Natural Science Foundation of China (52108259), and Aalborg University (CLAAUDIA).

## Author contributions

Z.C., T.D., and M.M.S. conceived the study and planned the computational work. Z.C. performed the molecular dynamics simulations, structural and dynamical analyses, and machine-learning calculations with input and assistance from T.D., N.M.A.K., Y.Y., and M.M.S. T.D. trained the neural-network potential. M.M.S. supervised the study. All authors participated in discussing the data and contributed to the writing of the manuscript.

## Competing interests

The authors declare no competing interests.
