## [Transparent Peer Review file · Nature Communications]

Disorder-induced enhancement of lithium-ion transport in solid-state electrolytes

Corresponding Author: Professor Morten Smedskjaer

Version 0:

Reviewer comments:

Reviewer #1

(Remarks to the Author)

1. The LPS was not considered in the MLIP training dataset. Given that many of the MD runs and analyses were performed at high temperatures, authors should check the accuracy of the fitted MLIP in terms of reproducibility of the high-temperature -phase of LPS. In fact, authors should include -LPS structures as part of the training dataset.
2. GGA-PBE overestimates lattice parameters which can largely affect diffusivity and activation barrier values. Authors should perform an objective comparison on GGA functional choice such as PBEsol with vdW, optB88-vdW, etc., to determine which of them are the most accurate vs. experimental results. To comprehensively evaluate the accuracy of the MLIP, authors should collect various experimental properties for comparison such as lattice parameters, ionic conductivities, activation energies, etc. from various measurement techniques (e.g., tof-sims, impedance spectroscopy, nmr, etc.) and discuss which one should be used for direct comparison and how these numerical values relate to the present calculation results based on the type of measured diffusivity data. These are aside from the structure factors and $g(r)$ data that were already analyzed in the present manuscript.
3. Comparison vs. already reported classical force-field potentials and MLIPs on LPS solid electrolytes should be discussed as well in order for the journal readers to make an assessment and analysis related to benchmarking and on the improvements being made for the present work.
4. Explicit grain boundary structures and related solid-solid interfaces (i.e., crystal-amorphous) were excluded in the training dataset, this results to uncertainty on the accuracy and robustness of the trained MLIP on whether it has already such structures based on the training dataset. Authors should show that the grain boundary and solid interface energies are accurately captured vs. DFT results.
5. In the description, it was highlighted that the descriptor is actually related to Li ion mobility and square of Li ion displacements (which is already a well-known idea, that is, the MSD plot/data). The way softness was used in the present manuscript can actually be simplified as well to the degree of structure disorder. Then, for the sake of clarity, why not used Li ion mobility or structure disorder descriptor instead which is a clearer description for a wider audience of journal readers, especially experimentalists? Why make such a distinction? is usually reserved for a materials mechanical property (e.g., sulfide-type vs. oxide-type solid electrolytes). Authors should further explain and clarify why the distinction of softness descriptor is necessary in this case as an appropriate term to use for defining such a feature related to Li ion dynamics, and why the Steinhardt order parameter is insufficient given the similarity in information that the latter provides.
6. Authors should clearly explain the origin of the discrepancies observed in Fig. S1 for the low-Q-region $S(Q)$ peaks and valleys.
7. In the error loss function for MLIP fitting, what were the weights used for the DFT energy, DFT force, and DFT stress tensor? Authors should provide this information.

Reviewer #2

(Remarks to the Author)

This work presents a profound and insightful theoretical exploration of solid electrolytes for batteries, offering valuable insights into their atomic-scale design. However, the structure of the paper, narrative flow, and presentation of data require substantial improvement to effectively convey the complex findings. While the data presented is intriguing, the format of a letter journal may not adequately accommodate the depth of analysis and discussion needed. A more extended format in a

specialized journal could better elucidate the results and their implications. A number of more in-depth comments are provided below to address these concerns and offer suggestions for improvement:

1. The glass-ceramic electrolyte system is introduced in the initial sections of the paper; however, a comprehensive understanding of this model is not fully elucidated until much later in the text. It is essential to provide a clearer and earlier description of this model and its significance within the study. Furthermore, the introduction mentions that the research focuses on quantifying the dynamics between internal order and disorder, but this concept lacks sufficient clarity. Similarly, the inclusion of interfaces is stated without a precise definition. Are these interfaces referring to grain boundaries or crystalline-glassy contacts? If so, how are these interfaces defined considering the expected complexity? Providing explicit definitions and explanations earlier in the paper would enhance the reader's understanding and appreciation of the study's objectives and methodologies.

2. A more comprehensive briefing on the structures associated with both crystalline beta and glassy Li₃PS₄ phases may be beneficial. While Figure 1b provides some insight, it is not entirely self-explanatory per se, making it challenging to fully grasp the underlying geometry of these phases and their distinctions between ordered, disordered, and glassy-like states. Clarification regarding the differences in structural arrangements, such as the degree of atomic arrangement regularity, bonding motifs, and density distributions, would enhance understanding. Specifically, elucidating the characteristics that differentiate ordered crystalline phases from disordered or glassy-like structures would provide valuable context for comparing and contrasting these states.

3. The authors assert the absence of previous long-term MD simulations available for describing glassy LiPS systems. However, it is noteworthy that in a recent paper, MLIPs were developed precisely for this purpose (<https://doi.org/10.1021/acsami.4c00618>). It may be relevant to cite and appropriately discuss this relevant work in the context of the current study.

4. While it is acknowledged in the introductions that classical force fields for LiPS systems lack the capacity to model reactivity, the assertion that the newly developed MLIP addresses this limitation may be misleading. The efficacy of the MLIP in addressing reactivity issues is not clearly elucidated. Although the training set incorporates diverse amorphous geometries, which indeed exhibit a wide range of bond distances and species, it does not adequately capture the dynamic evolution of reactivity scenarios. Consequently, crucial aspects of reactivity dynamics, especially those pertaining to contact with Li metal, are not comprehensively considered during the training process. As a result, the MLIP may not possess the capability to effectively learn and model such intricate reactivity phenomena.

5. At the end of the introduction, the paper asserts that the MLIPs developed herein, along with the accompanying MD simulations, holds broad applicability for studying ion transport in other solid-state systems. While it is acknowledged that training a robust MLIP can enhance both the time and length scales of MD simulations, the specific methodological advancements beyond this assertion remain unclear. Although the paper exemplifies the application of this approach, it falls short of delineating additional methodological innovations to substantiate such a claim. The introduction of the 'softness' fingerprint for classifying disorder-induced Li-ion diffusion, although previously introduced by the authors, is mentioned as a potential advancement. However, if the focus is on the MLIP development itself, further rigor in its generation and validation is warranted. Notably, training towards reactivity, a potential avenue for advancement, appears to be lacking. Consequently, as mentioned in my previous point too, the current MLIP may not be suitable for studying, for example, reactivity against Li metal, as it solely encompasses bulk phases without the inclusion of actual electrolyte-electrode AIMD simulations. As a result, the impact of the MLIP may be significantly minimized.

6. On page 3, the statement We find no regular coordination sites and symmetric long-range migration pathways within the glassy Li₃PS₄ electrolyte [7], implying isotropic ionic conduction [19] requires clarification regarding how this conclusion was reached from the simulations. Was it inferred solely through visual inspection or were there specific analytical techniques employed? Providing further details on the methodology or analysis techniques used to arrive at this conclusion would enhance clarity. Furthermore, a few lines below this statement, the concepts of octahedral and tetrahedral interstitial sites for Li ions are introduced without delving into their structural features or their significance in understanding the distinctions between ordered/crystalline and glassy structures. Elaborating on the characteristics and implications of these interstitial sites would aid in comprehending the structural differences and their influence on the behavior of Li ions within different phases.

7. On page 5, the statement The glassy, crystalline, and glass-ceramic Li₃PS₄ are three solid-state electrolytes with distinctly different structures, i.e., varying degrees of disorder may benefit from further clarification. It would be good to provide a more concrete understanding of how these structural differences are discerned and quantitatively characterized in terms of the degree of disorder. Specifically, elucidating the criteria or metrics used to define and quantify disorder within these electrolyte systems would enhance comprehension. Furthermore, expounding on the implications of these structural distinctions for classifying systems into glassy, crystalline, and glass-ceramic phases would provide valuable context for understanding their respective properties and behaviors.

8. In general, from line 124 to 148 on page 5, the text lacks clarity and could benefit from expanded explanations to improve the flow. For instance, the statement in reciprocal space, the beta-Li₃PS₄ exhibits a peak at lower Q-values compared to that in the glassy Li₃PS₄, signifying that its structural order extends over a longer length scale than that of glassy Li₃PS₄ may not be immediately clear to non-experts and requires further elaboration. Specifically, it could be helpful to explain how the location of peaks in reciprocal space reflects the length scale of structural order and why this comparison is significant. Additionally, in line 137-138, when it's mentioned that the distance distributions of Li and S remain largely consistent, it

might be clearer to use the term “largely similar instead of largely consistent to indicate that there are minimal variations between the distributions. And regarding the inference that “the Li atoms in the four-coordinated configuration occupy a larger free volume in the disordered structure as mentioned in line 140, it would be beneficial to explain how this conclusion was drawn from the data presented. Moreover, the discussion spanning these two paragraphs could be better organized to facilitate comprehension. Perhaps restructuring the discussion to follow a more logical progression and reducing the back-and-forth references between the main manuscript and supporting information would improve clarity. Ultimately, it seems that the key point of this section is that tracking the P-P distances (i.e., P2S6 and P2S7 species) serves as a descriptor for detecting glass disorder. Clarifying and emphasizing this key point throughout the discussion would help reinforce its significance and improve overall comprehension.

9. The results presented on pages 5-6 regarding the dynamics of hopping ions feature MD simulations conducted at very high temperatures (700-1000 K), presumably meant to accelerate the dynamics and facilitate ion diffusion at shorter time scales. While these temperatures are significantly higher than the operating conditions of these systems as electrolytes in batteries, it's important to note that MLIPs can easily facilitate long MD runs at room temperature conditions once created. This flexibility is indeed one of the advantages of MLIPs over AIMD. Therefore, it may be worth considering conducting simulations at lower temperatures more representative of real-world operating conditions to better capture the behavior of these systems in practical applications. Additionally, in the subsequent discussion, the statement Interestingly, the disordered structure, characterized by isotropic transport pathways, exhibits a smaller decrease in MSD with decreasing temperature compared to the more ordered structure lacks clear explanation of the connection between disorder, isotropic transport, and the network of connected octahedral and tetrahedral sites. It would be beneficial to provide a more detailed explanation of how these factors are interrelated and influence the observed trends. Specifically, elucidating how the structural characteristics of the disordered phase, such as isotropic transport pathways, contribute to its behavior under changing temperature conditions, and contrasting this with the behavior of the more ordered structure, would enhance understanding.

10. The discussion around Figures 2f and 2g presents a challenge in following the narrative coherently. There's a discrepancy between the temperatures mentioned in different sections; while some computations are conducted at very high temperatures (>700 K), others are performed at room temperature. This inconsistency makes it difficult to track the flow of the discussion. Additionally, the transition from discussing MSD and diffusion coefficients at high temperatures to ionic conductivities at room temperature, as depicted in Figure 2g, further complicates the understanding. The observation that the conductivity of the crystalline phase is higher than that of the glass and glass-ceramic phases in Figure 2g seems contradictory when considering the context of the high-temperature simulations discussed earlier. Moreover, the temperature range considered in Figure S8, spanning from 300 K to 700 K, lacks guidance or justification, adding to the confusion.

11. Furthermore, the sudden shift in nomenclature in Figure 2e, from using beta/g/gc_subindex_temperature labels to referring directly to the beta, glassy, and glass-ceramic phases, as consistently mentioned throughout the manuscript, adds to the difficulty in following the discussion seamlessly. Maintaining consistency in nomenclature throughout the figures and text would be desirable.

12. At the beginning of page 8, the concept of anomalous diffusion is introduced without initial elaboration. It might be more straightforward to initially introduce the concept of dynamic heterogeneity, as it is later described and referenced from the literature. This approach would simplify the text and improve readability by avoiding unnecessary complexity. Additionally, the non-Gaussian parameter mentioned in line 184 on page 8 is not defined or described anywhere in the text, further complicating the understanding of the analysis. Without proper explanation, it becomes challenging to comprehend how from Figure 2b one can infer that the motion of lithium ions exhibits non-Gaussian properties” as stated in line 188. While the position and height of the peak may seem relevant, readers are not informed about the significance or interpretation of these data points. Providing clear guidance on how to interpret these findings would enhance the flow of the discussion and make it easier to follow without needing to refer to the original paper.

13. The discussion on page 8, lines 205-207, regarding the orientation of the crystalline phases and the preferential diffusion path along the c-direction on the ac-plane in beta-Li3PS4 appears disconnected from the observation that the glass-ceramic phase does not exhibit intermediate properties between the beta and glass phases. This raises questions about the sensitivity of the results to the orientation of the ceramic-glass system and why a specific interface orientation was chosen over others. Exploring alternative interface orientations and discussing their potential impact on the results could provide valuable insights into the robustness of the findings. Furthermore, the statement on page 8, lines 208-209, that the z-direction is superior to that along the y-direction and significantly outperforms the x-direction” regarding diffusion requires more detailed discussion and analysis of the diffusion mechanisms involved. Simply describing the results without delving into the underlying mechanisms limits the depth of understanding.

14. The discussion on page 9, spanning lines 247 to 252, presents a challenge in readability as it involves referencing and comparing five different graphs from both the main manuscript and supplementary information. This dense concentration of information makes it difficult for readers to follow the discussion seamlessly. To enhance readability, it may be beneficial to streamline the discussion by consolidating the comparisons and providing concise summaries of the key findings from each graph. Perhaps, including visual aids such as tables summarizing the relevant data could assist readers in comprehending the comparisons without needing to constantly refer back and forth between multiple figures.

15. The final discussion provided on page 13 (lines 351-364) feels somewhat inadequate given the extensive volume of data presented throughout the paper. This raises questions about the suitability of the journal format for effectively conveying such complex findings. Moreover, the apparent conclusions drawn from the discussion lack generalization, leaving gaps in

understanding. With the complexity of the analyses, it remains unclear how disorder in the glass-ceramic system enhances ionic conductivity at low temperatures. It's possible that this enhancement is merely a consequence of the intrinsic lower activation energy of the present disordered phase. The discussion would benefit from further exploration of the role of Li-ion diffusion through the interface between the glass and crystalline phases in the glass-ceramic model. If the interface is thought to block Li-ion diffusion, then how is percolation of Li-ion observed in the study? This discrepancy is puzzling and warrants a more thorough examination and explanation. Overall, a deeper analysis and synthesis of the findings are needed to provide a more comprehensive and generalizable conclusion.

16. In the methods section, the choice of performing AIMD simulations at very high 3000 K raises concerns about its representativeness of actual glassy structures. The authors should provide justification for this procedure and compare it with alternative approaches. It's essential to ensure that the sampling is sufficiently robust, universal, and representative of the actual energy landscapes encountered in the formation of glassy or disordered structures. The selection of such a high temperature and short MD run duration (2.5 ps) may lead to a bias towards exploring configurations with very high energies, potentially deviating from the energy landscape of actual glassy structures. There's a risk of overlooking important configurations and dynamics that are crucial for understanding the formation of glassy or disordered structures. Therefore, the authors should thoroughly justify their choice of simulation parameters and demonstrate how they ensure the sampling captures the relevant energy landscape effectively.

17. The validation of the MLIP presented in Figure S1, particularly regarding the neutron structure factor, is not entirely satisfactory. There appears to be a discrepancy in both the position and relative intensities of the two main peaks in the range between 0 and 2.5 Q when compared to experimental data. Specifically, the simulated data is right-shifted with respect to the experiment, and the relative intensities of the peaks do not align as expected. Moreover, in Fig 1a, it's observed that the agreement between the MLIP and AIMD starts to deviate in the medium to long range, particularly beyond 3 Angstroms. In many cases, MLIPs are designed to capture short-range interactions accurately, as these typically dominate atomic-scale processes such as bonding and structure. Therefore, minor discrepancies in the medium to long range may not necessarily be alarming, especially if the MLIP accurately reproduces the essential features of the system at shorter distances. However, if the glassy structure is sensitive to long-range interactions, or if the deviations in the medium to long range become more pronounced with increasing distance, then this discrepancy may warrant further investigation. It's essential to assess the impact of these deviations on the overall accuracy and predictive capability of the MLIP.

18. In the methods section, it would be beneficial to provide commentary on the level of convergence achieved by the employed basis sets. This information is important for ensuring the reliability and accuracy of the results obtained from the simulations.

19. And again, in the methods section, it is relevant to include information on dataset size convergence during the training procedure. Ensuring dataset size convergence provides confidence that the diversity of the dataset structures has been adequately captured. This convergence assessment could involve monitoring the changes in model performance metrics, such as accuracy or error, as the dataset size increases.

20. How have been obtained the number of nearest neighbors per atom in Equation 7? Using Voronoi partition perhaps? Please clarify. And please, clarify as well if the redefinition of Equation 7 as Equation 8 aims to make the Steinhardt order parameter rotationally invariant or it has other motivations. Perhaps it's also worth considering alternative variants of Steinhardt order parameters, such as weighted versions [<https://doi.org/10.1063/1.4774084>] or w_l indexes [<https://doi.org/10.1103/PhysRevB.28.784>]. These variants may offer additional insights or advantages.

Other minor issues:

21. In the caption of Figure 1e, for consistency, it would be helpful to state that the solid line represents the RDFs and the dashed line represents the integrated RDF.

22. In page 6, line 162, referring to Figs. S5d-e appears to be a typographical error. It should likely refer to Figs. S5a-c instead.

23. Regarding the discussion of Figure S5 on page 6, it may be clearer to combine the data for the three systems (beta, glass, glass-ceramic) into a single figure with the three curves at 700 K. This would provide a more direct comparison and facilitate understanding.

24. The considered temperature in Figure 4a and S11 should be explicitly stated in the figure captions or the surrounding text to ensure clarity.

25. The concept of "Voronoi volume of lithium," introduced in the caption of Figure S4, should be described in the text.

26. The fraction of crystallinity presented in Figure S10 should be introduced and defined more smoothly in the main text.

Version 1:

Reviewer comments:

Reviewer #1

(Remarks to the Author)

Authors have addressed satisfactorily the comments that were raised.

Reviewer #2

(Remarks to the Author)

The authors have successfully addressed all previous concerns and the manuscript, in my opinion, is acceptable for publication.

REVIEWER COMMENTS

We thank the Editor and two Reviewers for their detailed and helpful comments concerning our manuscript. We have made point-by-point responses below and highlighted corrections in the main text and Supplementary information in red font.

Reviewer #1 (Remarks to the Author):

1. The α -LPS was not considered in the MLIP training dataset. Given that many of the MD runs and analyses were performed at high temperatures, authors should check the accuracy of the fitted MLIP in terms of reproducibility of the high-temperature α -phase of LPS. In fact, authors should include α -LPS structures as part of the training dataset.

Response: First, we thank the Reviewer for taking the time to review our manuscript and for the helpful comments and suggestions.

As you suggested, we have tested the accuracy of the present MLIP regarding the α -Li₃PS₄ phase. As shown in Response Fig. R1a below, the root-mean-square error (RMSE) of force prediction of α -Li₃PS₄ during 3000 K relaxation is 0.23 eV/Å. Following your suggestion, we have now included the α -Li₃PS₄ structures in the training dataset. The prediction results shown in Response Fig. R1b indicate that the performance of the MLIP is slightly improved (from 0.23 to 0.21 eV/Å) upon including it in the training dataset. This indicates that the MLIP already contained the majority information regarding the local atomic structure in the α -Li₃PS₄ phase, even though the phase was not explicitly included in the original training set. Accordingly, we have updated the results in the revised Supplementary Information as well as updated the machine learning potential file in the GitHub repository (<https://github.com/OxideGlassGroupAAU/LiPS>).

Response Fig. R1. Comparison of MLIP predicted and DFT calculated atomic forces for α -Li₃PS₄ phase using the MLIP trained (a) without and (b) with α -Li₃PS₄ datasets.

2. GGA-PBE overestimates lattice parameters which can largely affect diffusivity and activation barrier values. Authors should perform an objective comparison on GGA functional choice such as PBEsol with vdW, optB88-vdW, etc., to determine which of them are the most accurate vs. experimental results. To comprehensively evaluate the accuracy of the MLIP, authors should collect various experimental properties for comparison such as lattice parameters, ionic conductivities, activation energies, etc. from various measurement techniques (e.g., tof-sims, impedance spectroscopy, nmr, etc.) and discuss which one should be the used for direct comparison and how these numerical values relate to the present calculation results based on the type

of measured diffusivity data. These are aside from the structure factors and $g(r)$ data that were already analyzed in the present manuscript.

Response: Indeed, we have used the GGA-PBE with DFT-D3 vdW corrections to generate the training datasets. We acknowledge that more accurate functionals exist that can better describe the structure and properties of LiPS systems but come with much higher computational costs such as meta-GGA and hybrid-GGA. To address the Reviewer's concern, we have compared the MD simulated structures of β -Li₃PS₄ using the MLIPs trained from different datasets, i.e., the datasets generated from DFT calculations with PBE-vdW (this work) and PBEsol-vdW functionals (the dataset is from Wang et al., <http://arxiv.org/abs/2406.18263>), respectively. As shown in Response Fig. R2, both MLIPs can reproduce the main features of the crystalline β -Li₃PS₄ structure, i.e., the PS₄ tetrahedra are periodically aligned with Li ions inserted in between.

Response Fig. R2. Supercell structure ($8\times 8\times 4$) of β -Li₃PS₄ crystal after relaxing at 300 K and zero pressure using the MLIP trained from the DFT datasets based on (a) PBE-vdW, and (b) PBEsol-vdW.

Based on the simulated trajectories, we have also calculated the pair distribution functions to quantify the differences in short-range structures. As shown in Response Fig. R3, the two structures exhibit a similar structure regarding the PS₄ tetrahedra except for the Li-Li correlations.

Response Fig. R3. Pair distribution functions of (a) total, (b) Li-Li, (c) Li-S, and (d) P-S atom pairs of the β - Li_3PS_4 crystal simulated using MLIP trained based on PBE-vdW and PBEsol-vdW datasets.

Next, we compare the structure of glassy β - Li_3PS_4 using the melt-quenching method with two different MLIPs. Unfortunately, the PBEsol-vdW MLIP cannot be applied in the high-temperature simulation as it introduces an unphysical agglomeration at 1500 K (Response Fig. R4a). This is also reflected by the peak of 2.0 Å in the S-S partial pair distribution function (Response Fig. R4b). Therefore, the temperature to melt β - Li_3PS_4 was limited to 1200 K for the PBE-sol-vdW MLIP. Response Fig. R4c shows a comparison of simulated glass structures, including experimental data of the glass structure from Garcia-Mendez et al. (Adv. Energy Mater. 10, 2000335 (2020)). The simulated glass structure of PBE-vdW MLIP exhibits a better agreement with the reference, which further confirms the rationality of using PBE-vdW MLIP for the present study.

Response Fig. R4. (a) Schematic snapshot of the unphysical agglomeration within Li_3PS_4 at 1500 K simulated using the PBEsol-vdW MLIP. (b) S-S partial pair distribution of Li_3PS_4 simulated at different temperatures using two MLIPs. (c) Li-S partial pair distribution of simulated Li_3PS_4 glass using different MLIPs. The experimental reference data is from Garcia-Mendez et al. (*Adv. Energy Mater.* 10, 2000335 (2020)).

Finally, we have also added the comparison between the present simulation results and various experimental properties. We have summarized these results in Response Table R1 (shown as Table S1 in the revised Supplementary Information). Overall, our MLIP exhibits excellent agreement with experimental results, both in terms of structural reproduction (see Response Fig. R5 below in response to your comment no. 6) and ionic conductivity and activation energy. We have added all comparison and discussion in the revised manuscript and Supplementary Information.

Response Table R1. Lattice parameters, room temperature ionic conductivity (σ_{RT}), and conduction activation energy (E_a) of Li_3PS_4 electrolytes. Data are included from this work as well as literature, covering both experimental results (based on x-ray diffraction) as well as simulation results using different methods. The abbreviations used in the table are as follows: Exp. (experiment), DFT (density functional theory), CMD (classical molecular dynamics), and MTP (moment tensor potential).

Li_3PS_4	Lattice parameters			σ_{RT} (S cm ⁻¹)	E_a (eV)
	a (Å)	b (Å)	c (Å)		
β - Li_3PS_4 (this work)	13.01	8.34	6.22	4.8×10^{-6}	0.49
Exp. ¹	12.82	8.22	6.12	8.9×10^{-7}	0.48 ²
Exp. ³	12.98	8.04	6.13		
DFT based on PBE ⁴	13.07	8.13	6.26		
DFT based on PBE ⁵	13.02	8.17	6.25	6×10^{-5}	0.49
DFT based on PBE (this work)	13.03	8.02	6.17		
DFT based on PBEsol (this work)	12.85	7.93	6.10		
DFT based on PBEsol ⁶				7.7×10^{-4}	0.38
DFT based on PBE0 ⁶				8.7×10^{-6}	0.62
CMD ⁷				$\sim 10^{-2}$	<0.2
MTP ⁸	13.06	8.12	6.26	7.4×10^{-5}	0.29
Li_3PS_4 glass (this work)				1.3×10^{-4}	
Exp. ⁹				1.8×10^{-4}	
Li_3PS_4 glass-ceramic (this work)				2.2×10^{-4}	
Exp. ⁹				2.8×10^{-4}	

3. Comparison vs. already reported classical force-field potentials and MLIPs on LPS solid electrolytes should be discussed as well in order for the journal readers to make an assessment and analysis related to benchmarking and on the improvements being made for the present work.

Response: Good point. We have added discussion in the revised manuscript and Supplementary Information comparing the present MLIP with classical potentials. Specifically, we discussed and compared the accuracy of classical molecular dynamics (CMD) force field⁷ and other MLIPs^{8,10} with the present MLIP in calculating lattice parameters, ionic conductivity, and activation energy. As stated in our response to your comment no. 2 above, the results in Response Table R1 shows that our MLIP provides more accurate calculations compared to the CMD and recent MLIP for the lattice parameters, ionic conductivity, and activation energy of β - Li_3PS_4 crystals, glass, and glass ceramics. In detail, the existing CMD⁷ and other MLIPs^{8,10} tend to overestimate the ionic conductivity and activation energy of β - Li_3PS_4 crystals and Li_3PS_4 glass. Additionally, we have compared the accuracy of other recent MLIPs with our MLIP in reproducing glass structures, particularly in terms of the structure factor. That is, we have included comparison to recently published MLIP on LiPS solid electrolytes (as presented in Response Fig. R5). As shown in Response Fig. R5, our MLIP matches the experimental values better in both the peak position and intensity on neutron structure factor ($S_N(Q)$) compared to other MLIPs. We have discussed the above content in detail in the first subsection of the results section entitled “Machine learning interatomic potential” and in Table S1 of the Supplementary Information.

Response Fig. R5. Comparison of simulated and experimental neutron structure factor $S_N(Q)$. Experimental and other MLIP-calculated $S_N(Q)$ were obtained from the Refs^{10,11}.

4. Explicit grain boundary structures and related solid-solid interfaces (i.e., crystal-amorphous) were excluded in the training dataset, this results to uncertainty on the accuracy and robustness of the trained MLIP on whether it has already “seen” such structures based on the training dataset. Authors should show that the grain boundary and solid interface energies are accurately captured vs. DFT results.

Response: We agree that this is important to check that the grain boundary and solid interface energies are accurately captured by the MLIP. As mentioned in the Methods section, our training datasets include the melting process of crystalline phases at 3000 K, which based on our experience should allow the MLIP to capture the crystalline-to-amorphous phase transition. Following the reviewer’s suggestion, we have further tested the ability of our MLIP to predict the interfacial structure. We built the interfacial structure by partially melting the upper part of a β - Li_3PS_4 crystal at 3000 K for 2.5 ps using *ab initio* MD simulations (see Response Fig. R6a). The resulting interfacial structure was then equilibrated at 1000 K to sample the energy, force, and virial information. As shown in Response Fig. R6b, the force prediction of the interfacial structure exhibits a comparable accuracy compared with the structure already contained in the training sets (as also highlighted in Fig. S21 of the revised Supplementary Information). Therefore, we conclude that the accuracy and robustness of the present MLIP is sufficient to be used for simulating the grain boundary structures as well as the crystal-amorphous interfaces.

Response Fig. R6. (a) Atomic structure of the interface between the β - Li_3PS_4 crystal and amorphous structures used for *ab initio* MD simulations. Lithium atoms are colored in brown, sulfur atoms in blue, and phosphorus atoms in yellow. (b) Comparison of atomic forces from MLIP predictions and DFT calculations for the crystal-amorphous structure of β - Li_3PS_4 . Here the MLIP was trained without the interfacial structure in the dataset.

5. In the description, it was highlighted that the “softness” descriptor is actually related to Li ion mobility and square of Li ion displacements (which is already a well-known idea, that is, the MSD plot/data). The way “softness” was used in the present manuscript can actually be simplified as well to the degree of structure disorder. Then, for the sake of clarity, why not used “Li ion mobility” or “structure disorder” descriptor instead which is a clearer description for a wider audience of journal readers, especially experimentalists? Why make such a distinction? “Softness” is usually reserved for a material’s mechanical property (e.g., sulfide-type vs. oxide-type solid electrolytes). Authors should further explain and clarify why the distinction of “softness” descriptor is necessary in this case as an appropriate term to use for defining such a feature related to Li ion dynamics, and why the Steinhardt order parameter is insufficient given the similarity in information that the latter provides.

Response: We acknowledge that more explanation about the “softness” descriptor would be helpful for readers. First, we agree with the Reviewer that the “softness” descriptor in this work is correlated with the degree of structure order, but this does not mean that “softness” can be simplified as the degree of structural order. The main challenge of understanding disordered materials comes from the complex structure at different length scales. As we do not know all the possible descriptors, we use machine learning to help find such advanced descriptor of the disordered structures.

In detail, the calculation of “softness” is not related to mechanical properties but relies on the pairwise radial correlation functions and angular distribution functions, which contains high dimensional information of the local environment of Li ions. The term “softness” was introduced in the original work on this metric (by Cubuk et al.^{12,13}) and hence we use it here for the sake of consistency. Specifically, the softness is a function of the distance from the hyperplane classifying mobile and immobile particles ($\sum_r W(r)G(k; \mu) - b = 0$) based on their radial distribution function ($G(k; \mu) = \sum_i e^{-(r_{ik}-\mu)^2/L^2}$). Here, $W(r)$ and b are the weight coefficients and bias of the logistic regression model, respectively. The hyperplane is a linear combination of input features, allowing softness to be determined based on different features. In other words, the absolute value of $W(r)$ indicates the importance of the corresponding feature $G(k, \mu)$, with positive and negative signs signifying that an increase in the value of $G(k, \mu)$ will, respectively, increase or decrease the softness value. In turn, these terms include the information of structure disorder.

During the training process, the machine learning algorithm automatically finds the most significant structural features that correlate with the designated properties, i.e., Li^+ mobility in this work. Coincidentally, the calculated “softness” is correlated with the degree of structure disorder, indicating the significance of structure disorder in governing Li^+ mobility. We have added a description of the term “softness” in the Methods section on softness of the revised manuscript, detailing its purpose and implementation results. As it follows from the description above, the implementation of softness is based on the local structure and the rearrangement capability of atoms. It can predict the dynamic properties of atoms solely from the static structure, such as lithium migration capabilities in different local environments. Given the complexity of local structures in disordered systems, it is impossible to identify all potential descriptors. We believe that the use of machine learning significantly accelerates and simplifies this process.

6. Authors should clearly explain the origin of the discrepancies observed in Fig. S1 for the low-Q-region $S(Q)$ peaks and valleys.

Response: Thank you for pointing this out. The differences in the structure factor ($S_M(Q)$) between experiment and simulation were due to our calculation error, specifically, the atomic numbers and neutron scattering lengths were incorrectly assigned during the calculations. We have recalculated $S_M(Q)$ and ensured the accuracy of the results. Using the example in our GitHub repository (<https://github.com/OxideGlassGroupAAU/LIPS>), this result can be accurately reproduced. For comparison, we present in Response Fig. R5 above the comparison between the $S_M(Q)$ of Li_3PS_4 glass calculated using the present MLIP and the experimental results¹¹, showing very good agreement with Wright factor R_x of 4.6% (Wright, J. Non-Cryst. Solids 159, 264 (1993)). In the same figure, we also compare the $S_M(Q)$ results calculated using another recent MLIP¹⁰, which also shows a good agreement (although slightly worse than the present MLIP, with Wright factor R_x of 5.3%).

7. In the error loss function for MLIP fitting, what were the weights used for the DFT energy, DFT force, and DFT stress tensor? Authors should provide this information.

Response: We have provided this information in the Methods section of the revised manuscript. Specifically, during the initial training process, the weight of the DFT energy, force, and virial terms were dynamically changed from 0.02 to 2, 1000 to 1, and 0.02 to 0.2, respectively. During the second training process, the weight of the DFT energy, force, and virial terms were kept at 1.

Reviewer #2 (Remarks to the Author):

This work presents a profound and insightful theoretical exploration of solid electrolytes for batteries, offering valuable insights into their atomic-scale design. However, the structure of the paper, narrative flow, and presentation of data require substantial improvement to effectively convey the complex findings. While the data presented is intriguing, the format of a letter journal may not adequately accommodate the depth of analysis and discussion needed. A more extended format in a specialized journal could better elucidate the results and their implications. A number of more in-depth comments are provided below to address these concerns and offer suggestions for improvement:

Response: We thank the Reviewer for taking the time to review our manuscript and for all their helpful comments, which are addressed point-by-point in the following. As confirmed by the Editor, the present journal indeed allows for full-length research papers. Consequently, we have expanded the manuscript with more details and fully rewritten several sections to improve the clarity of presentation.

1. The glass-ceramic electrolyte system is introduced in the initial sections of the paper; however, a comprehensive understanding of this model is not fully elucidated until much later in the text. It is essential to provide a clearer and earlier description of this model and its significance within the study. Furthermore, the introduction mentions that the research focuses on quantifying the dynamics between internal order and disorder, but this concept lacks sufficient clarity. Similarly, the inclusion of interfaces is stated without a precise definition. Are these interfaces referring to grain boundaries or crystalline-glassy contacts? If so, how are these interfaces defined considering the expected complexity? Providing explicit definitions and explanations earlier in the paper would enhance the reader's understanding and appreciation of the study's objectives and methodologies.

Response: Thank you for your comments on improving the paper. As you suggested, we have added some background information about glass-ceramics and interfaces in the Introduction section of the revised manuscript. In this section, we have also revised the explanation regarding the quantification of internal order and disorder. In the beginning of the Results section, we have defined the interfaces using amorphization quantification method as stated in the revised manuscript. In detail, we first introduce the definition and advantages of glass ceramics, followed by an overview of the current research status on glass ceramic electrolytes. We then describe the complexity of glass ceramic structures, particularly the complex interfaces between the glassy and crystalline phases, which has received relatively little attention so far. Finally, we emphasize the role of this interfacial phase and conclude by stating that the complex structure of glass ceramics and the interfacial phase, as well as their impact on ion migration, require further investigation, which is what we have done in this work.

2. A more comprehensive briefing on the structures associated with both crystalline beta and glassy Li_3PS_4 phases may be beneficial. While Figure 1b provides some insight, it is not entirely self-explanatory per se, making it challenging to fully grasp the underlying geometry of these phases and their distinctions between ordered, disordered, and glassy-like states. Clarification regarding the differences in structural arrangements, such as the degree of atomic arrangement regularity, bonding motifs, and density distributions, would enhance understanding. Specifically, elucidating the characteristics that differentiate ordered crystalline phases from disordered or glassy-like structures would provide valuable context for comparing and contrasting these states.

Response: As suggested, we now provide more discussion after Fig. 1b on the characteristics of crystal structures and the differences between glass structures, etc. in the revised manuscript. These modifications were included after introducing the glass-ceramic model in Fig. 1c, as it helps us compare and discuss the differences between glass, crystalline, and glass-ceramic materials. We have also modified Fig. 1b to show only the difference in the structure for the crystal and glassy Li_3PS_4 . Regarding the relationship between structure and ion migration, we respond to this with more details below in relation to your Comment no. 6. We have thus added more information about the differences in atomic arrangement regularity and bonding motifs between glass and crystalline Li_3PS_4 in the revised manuscript.

Finally, we have also included atomic density distribution that visually illustrate the structural differences between ordered and disordered regions. This is shown in Response Fig. R7 below and as the new Supplementary Figure S3 in the revised Supplementary Information.

Response Fig. R7. Gaussian density distribution projections of P and S atoms in the 2D plane for (a) glass-ceramic Li_3PS_4 , (b) glassy Li_3PS_4 , and (c) $\beta\text{-Li}_3\text{PS}_4$ electrolytes.

3. The authors assert the absence of previous long-term MD simulations available for describing glassy LiPS systems. However, it is noteworthy that in a recent paper, MLIPs were developed precisely for this purpose (<https://doi.org/10.1021/acsami.4c00618>). It may be relevant to cite and appropriately discuss this relevant work in the context of the current study.

Response: Yes, this is a relevant paper to cite and discuss, thank you for pointing us to it. Furthermore, we note that some machine learning potentials for lithium thiophosphate electrolytes have emerged after the date we submitted this study. Recently, Huang et al.¹⁴ developed a machine learning potential for $\text{Li}_{10}\text{GeP}_2\text{S}_{12}$ type materials, but not focused on glassy electrolytes. However, to our knowledge, there is so far no MLIP for lithium thiophosphate electrolytes with both ordered and disordered structures.

To compare these different potentials and approaches, we have collected the relevant data and included the comparison and discussion in the revised manuscript and Supplementary Information. Specifically, in the beginning of the Results section in the revised manuscript, we have added a new subsection entitled “Machine learning interatomic potential.” In this subsection, we compare the ability of our MLIP to reproduce the structure of the Li_3PS_4 system with that of AIMD and CMD. Additionally, we provided a comparison of our MLIP with other MLIPs, as well as with DFT and experimental data. This is given in Table S1 of the revised Supplementary Information, where the relevant references are cited. Additionally, please also see our response to Comment no. 3 of Reviewer #1 above, which also is related to such comparisons (Response Table R1 and Response Fig. R5).

4. While it is acknowledged in the introductions that classical force fields for LiPS systems lack the capacity to model reactivity, the assertion that the newly developed MLIP addresses this limitation may be misleading. The efficacy of the MLIP in addressing reactivity issues is not clearly elucidated. Although the training set incorporates diverse amorphous geometries, which indeed exhibit a wide range of bond distances and species, it does not adequately capture the dynamic evolution of reactivity scenarios. Consequently, crucial aspects of reactivity dynamics, especially those pertaining to contact with Li metal, are not comprehensively

considered during the training process. As a result, the MLIP may not possess the capability to effectively learn and model such intricate reactivity phenomena.

Response: We acknowledge that our potential function lacks sufficient validation in dynamic evolution of reaction, although it incorporates the dynamics of chemical bond breaking and formation. To this end, we note that the use of machine learning-based potentials reduces the cost of simulations with accuracy close to *ab initio* by several orders of magnitude, but it still remains too expensive for the computation of reactions that go beyond bond breaking and re-forming, making it impossible to explore the timescales on which these reaction processes occur. This also makes the collection of reference configurations and the construction of interatomic potentials challenging. However, enhanced sampling techniques can not only bridge the timescale gap but also yield appropriate configuration sets for training potentials, allowing for DFT-quality reaction simulations of rare events that would otherwise be beyond the scope of classical and *ab initio* simulations¹⁵.

Based on the Reviewer's concern, we have revised our description regarding reactivity. It is important to emphasize that our potential development serves both glassy and crystalline lithium-phosphorus-sulfur family electrolytes. An accurate potential for describing interatomic interactions is a crucial prerequisite for studying the relationship between structure and performance. This paper focuses on modeling solid-state electrolytes of various degrees of order and investigates how disorder at the atomic scale facilitates lithium-ion transport, thereby explaining the performance observed in experiments and establishing an accurate link between electrolyte performance and atomic structure.

Importantly, although our potential has not been trained to capture reactions between lithium metal and the electrolyte, it has demonstrated the capability to capture chemical reactions within the structure. That is, the conversion between PS_4 , P_2S_6 , and P_2S_7 units in $Li_2S-P_2S_5$ glasses, as shown in Fig. 1g of the revised manuscript. Although the initial structure only consists of PS_4 units, there are considerable amount of P_2S_6 units existing in the glassy structure after melt-quenching. This highlights that our MLIP is able to capture the chemical reactions within the LiPS system, which have previously been experimentally observed^{11,16,17}.

5. At the end of the introduction, the paper asserts that the MLIPs developed herein, along with the accompanying MD simulations, holds broad applicability for studying ion transport in other solid-state systems. While it is acknowledged that training a robust MLIP can enhance both the time and length scales of MD simulations, the specific methodological advancements beyond this assertion remain unclear. Although the paper exemplifies the application of this approach, it falls short of delineating additional methodological innovations to substantiate such a claim. The introduction of the 'softness' fingerprint for classifying disorder-induced Li-ion diffusion, although previously introduced by the authors, is mentioned as a potential advancement. However, if the focus is on the MLIP development itself, further rigor in its generation and validation is warranted. Notably, training towards reactivity, a potential avenue for advancement, appears to be lacking. Consequently, as mentioned in my previous point too, the current MLIP may not be suitable for studying, for example, reactivity against Li metal, as it solely encompasses bulk phases without the inclusion of actual electrolyte-electrode AIMD simulations. As a result, the impact of the MLIP may be significantly minimized.

Response: Thank you for the comments. In fact, what we intended to express is that the entire set of methods used in this paper can be applied to the computational study of all ordered and disordered solid-state electrolytes at a fundamental research level. First, when investigating the structure-property relationships in solid-state electrolytes, it is crucial to accurately describe the interatomic interactions and replicate the structure. At the same time, studying ionic dynamics requires large-scale and long-time simulations, where DFT-based simulations cannot meet the demands for both high accuracy and efficiency, while classical potentials lack sufficient accuracy. MLIP strikes a balance between high accuracy and efficiency, making it a

strong tool for computational materials research. In this study, we have used MLIP to accurately reproduce the structure of the LiPS system, particularly the various combinations of disordered and ordered structures, which allowed us to further investigate lithium-ion diffusion behavior. Accordingly, we also ensure the generation and accuracy of MLIPs. In conjunction with our responses to Comments no. 1-4 from Reviewer #1, we have further validated the potential's accuracy in the revised manuscript.

An accurate potential function is the foundation for our subsequent studies on the structure-property relationships. For disordered solid-state electrolytes with more complex structures, which are among the most promising solid-state electrolytes, understanding the structural origin of ionic conduction behavior is crucial. However, due to the lack of long-range order, analyzing their structure becomes more challenging. To address this, we need a structural descriptor to uncover the hidden information within the structure. In this study, we used the "softness" metric, which is based on local structural information and applies machine learning to classify the information associated with different lithium atoms, allowing us to further analyze the structural origins of diffusion. In the revised manuscript, we have elaborated on the advantages of the softness method and provided a detailed response to Comment no. 5 of Reviewer #1 above.

Regarding MLIP's capture of chemical reactions, as we addressed in response to your Comment no. 4, the focus of the methodology and potential functions used in our article is not to capture electrolyte-electrode chemical reaction events but to provide accurate descriptions of structures and ion dynamics.

6. On page 3, the statement "We find no regular coordination sites and symmetric long-range migration pathways within the glassy Li₃PS₄ electrolyte [7], implying isotropic ionic conduction [19]" requires clarification regarding how this conclusion was reached from the simulations. Was it inferred solely through visual inspection or were there specific analytical techniques employed? Providing further details on the methodology or analysis techniques used to arrive at this conclusion would enhance clarity. Furthermore, a few lines below this statement, the concepts of octahedral and tetrahedral interstitial sites for Li ions are introduced without delving into their structural features or their significance in understanding the distinctions between ordered/crystalline and glassy structures. Elaborating on the characteristics and implications of these interstitial sites would aid in comprehending the structural differences and their influence on the behavior of Li ions within different phases.

Response: We have revised the description on p. 5 of the revised manuscript to enhance the clarity. Specifically, we have compared the structures of the crystalline and glassy phases, as shown in Response Figure R8 below (Supplementary Fig. S2 in the revised Supplementary Information). For the crystalline β -Li₃PS₄, the regularly arranged PS₄ units form the sites for lithium migration. Based on experimental and simulation results from the references^{1,18}, we have identified different migration sites and potential migration pathways and provided a discussion on these. However, in the glassy electrolyte, due to the disordered arrangement of the PS₄ units and the lack of regular coordination sites and symmetrical migration pathways (as described in the references^{19,20} for disordered structures), we are unable to define corresponding migration sites and pathways. We have also enhanced the discussion on this aspect in the revised manuscript.

Response Fig. R8. Atomic snapshots of (a,b) β - Li_3PS_4 and (c) glassy Li_3PS_4 electrolyte configurations. Li1 and Li2 are lithium sites located at the centers of Li-S tetrahedra and octahedra, respectively. Li3 is a tetrahedral interstitial site, while i2 and i3 are octahedral interstitial sites. The snapshots are captured from localized regions within the final relaxed configurations in the MD simulation using the present MLIP.

To address the second part of the comment, we have discussed the roles of tetrahedral and octahedral lithium sites, as well as interstitial sites, in the lithium-ion migration process within β - Li_3PS_4 crystals in more details in the revised manuscript and Supplementary Information. Supplementary Fig. S2 shows the configurations of β - Li_3PS_4 and glassy Li_3PS_4 electrolytes with the defined sites labeled. Additionally, we have explained in the relevant discussion sections related to Fig. 1e and Response Fig. R9 (reproduced as Supplementary Fig. S6 in the revised Supplementary Information) that the tetrahedral and octahedral sites correspond to different Li-S coordination states, specifically, four-coordinated and six-coordinated Li. To this end, we have now discussed the unique lithium sites and migration pathways in the crystal, formed by tetrahedral and octahedral coordination.

In the glassy electrolyte, lithium is characterized by tetrahedral coordination, which is also reflected in the difference in atomic volumes, as shown in Response Fig. R9 (with octahedrally coordinated lithium having a smaller atomic volume than tetrahedrally coordinated lithium). Due to the complex local structure of the glassy electrolyte, each tetrahedrally coordinated lithium site has a distinct local environment, making its migration pathways more complex, unlike the more regular paths formed by tetrahedral and octahedral lithium sites in the crystal. Following the Reviewer's suggestion, we have included further discussion on this in the revised manuscript.

Response Fig. R9. (a,b) Voronoi volume of lithium atoms in simulated Li_3PS_4 electrolytes using the MLIP: (a) density distribution and (b) violin plot. (c) Voronoi volume distribution of four-fold and six-fold coordination lithium atoms. (d) Averaged volume of four-fold coordinated lithium atoms in simulated glassy Li_3PS_4 , $\beta\text{-Li}_3\text{PS}_4$, and glass-ceramic Li_3PS_4 electrolytes.

7. On page 5, the statement “The glassy, crystalline, and glass-ceramic Li_3PS_4 are three solid-state electrolytes with distinctly different structures, i.e., varying degrees of disorder” may benefit from further clarification. It would be good to provide a more concrete understanding of how these structural differences are discerned and quantitatively characterized in terms of the “degree of disorder”. Specifically, elucidating the criteria or metrics used to define and quantify disorder within these electrolyte systems would enhance comprehension. Furthermore, expounding on the implications of these structural distinctions for classifying systems into glassy, crystalline, and glass-ceramic phases would provide valuable context for understanding their respective properties and behaviors.

Response: Good point. We have now noted in the revised manuscript that the degree of disorder in the bulk phase is measured based on crystallinity. Completely disordered glassy Li_3PS_4 and ordered $\beta\text{-Li}_3\text{PS}_4$ represent the extremes, with the glass-ceramics being partially crystalline. As the crystallinity of glass-ceramics increases, the sample’s overall degree of disorder decreases. Crystallinity is then calculated based on the fraction of the volume occupied by the crystalline phase relative to the total volume of the glass-ceramic. As

illustrated in Response Fig. R10, the crystalline phase occupies approximately 33% of the total volume, which is used to measure the crystallinity.

Response Fig. R10. Atomic snapshot of simulated glass-ceramic Li_3PS_4 , with annotated dimensions representing the sizes of the crystalline and glassy phases.

Glass-ceramics contain phases with different degree of disorder, including glassy phases, interfacial phases, and crystalline phases with a completely ordered structure. The internal disorder is quantified by calculating the amorphization factor $F(Z)$, as shown in Figure 1c in the manuscript. The specific calculation method for $F(Z)$ is detailed in the Methods section of the revised manuscript. It primarily quantifies the degree of atomic arrangement disorder. Specifically, we followed a procedure similar to that in ref²¹, beginning with a 3D Gaussian density distribution of the atomic positions. After mapping these positions onto a 3D grid, we created density slabs along the y -axis of the simulation box with a width of Δy and projected them into 2D. By summing the intensities, which were obtained through a 2D discrete Fourier Transform with the zero-frequency component shifted to the center, we then calculated $F(Z)$ by normalization to the maximum value:

$$F(Z) = \frac{\sum_{xz} I_{2D-FFT}(y)}{\sum_{xz} I_{max}}$$

8. In general, from line 124 to 148 on page 5, the text lacks clarity and could benefit from expanded explanations to improve the flow. For instance, the statement “in reciprocal space, the beta- Li_3PS_4 exhibits a peak at lower Q -values compared to that in the glassy Li_3PS_4 , signifying that its structural order extends over a longer length scale than that of glassy Li_3PS_4 ” may not be immediately clear to non-experts and requires further elaboration. Specifically, it could be helpful to explain how the location of peaks in reciprocal space reflects the length scale of structural order and why this comparison is significant. Additionally, in line 137-138, when it's mentioned that “the distance distributions of Li and S remain largely consistent”, it might be clearer to use the term “largely similar” instead of “largely consistent” to indicate that there are minimal variations between the distributions. And regarding the inference that “the Li atoms in the four-coordinated configuration occupy a larger free volume in the disordered structure”, as mentioned in line 140, it would be beneficial to explain how this conclusion was drawn from the data presented. Moreover, the discussion spanning these two paragraphs could be better organized to facilitate comprehension. Perhaps restructuring the discussion to follow a more logical progression and reducing the back-and-forth references between the main manuscript and supporting information would improve clarity. Ultimately, it seems that the key point of this section is that tracking the P-P distances (i.e., P2S6 and P2S7 species) serves as a descriptor for detecting glass disorder. Clarifying and emphasizing this key point throughout the discussion would help reinforce its significance and improve overall comprehension.

Response: Thank you for the helpful suggestions to improve readability. We have reorganized this section to make the discussion more logically coherent. We have also added more discussion on the relationship between different peak positions (Q -values) and the structure. As suggested, we have changed the term "largely consistent" to "largely similar" to convey the meaning more accurately.

Additionally, we have revised the data presentation of the atomic volumes in Fig. S6 of the revised Supplementary Information to ensure the conclusions can be intuitively drawn from the figure. The explanation of how the position of the structure factor peaks in reciprocal space reflects the length scale of structural order is now stated as follows: "Since the scattering vector Q is the inverse distance in real space, the low Q -values region of the $S(Q)$ encompasses structural information from SRO to MRO. Generally, different peak positions in the low Q -values region of $S(Q)$ correspond to ordering at different scales. The first peak represents the arrangement of motifs at medium range, the second peak reflects the size of local network-forming motifs, and the third peak provides information about nearest-neighbor interactions²². The first peak in the low Q -values region of $S(Q)$ is also known as the First Sharp Diffraction Peak (FSDP), which characterizes the topological arrangement of motifs under MRO. This feature varies among various disordered systems and its intensity is highly sensitive to the degree of disorder²²." The conclusion that four-fold coordinated Li atoms occupy a larger free volume in the disordered structure is visually demonstrated in Supplementary Figs. S6a-d of the revised Supplementary Information. Response Fig. R9 above is a reproduction of these figures.

9. The results presented on pages 5-6 regarding the dynamics of hopping ions feature MD simulations conducted at very high temperatures (700-1000 K), presumably meant to accelerate the dynamics and facilitate ion diffusion at shorter time scales. While these temperatures are significantly higher than the operating conditions of these systems as electrolytes in batteries, it's important to note that MLIPs can easily facilitate long MD runs at room temperature conditions once created. This flexibility is indeed one of the advantages of MLIPs over AIMD. Therefore, it may be worth considering conducting simulations at lower temperatures more representative of real-world operating conditions to better capture the behavior of these systems in practical applications. Additionally, in the subsequent discussion, the statement "Interestingly, the disordered structure, characterized by isotropic transport pathways, exhibits a smaller decrease in MSD with decreasing temperature compared to the more ordered structure" lacks clear explanation of the connection between disorder, isotropic transport, and the network of connected octahedral and tetrahedral sites. It would be beneficial to provide a more detailed explanation of how these factors are interrelated and influence the observed trends. Specifically, elucidating how the structural characteristics of the disordered phase, such as isotropic transport pathways, contribute to its behavior under changing temperature conditions, and contrasting this with the behavior of the more ordered structure, would enhance understanding.

Response: Good point. To address the comment, we have performed new simulations of lithium-ion diffusion at lower temperatures, ranging from 300 to 1000 K. It is important to note that as the temperature decreases, the movement of lithium ions gradually shifts away from the diffusive region, exhibiting sub-diffusion, i.e., deviation from Brownian motion, as shown in Response Fig. R11a. Therefore, due to the fact that the movement of particles under sub-diffusion is hindered by certain mechanisms (such as energy barriers, viscosity, etc.), the MSD ($\overline{\langle r^2(t) \rangle}$) increases sub-linearly with time t (i.e., $\langle r^2(t) \rangle \propto t^\alpha$, where $\alpha < 1$). As a result, the estimation of the diffusion coefficient is inaccurate. Our previous room temperature conductivity was extrapolated from high-temperature data, which generally results in higher values compared to simulations at room temperature, as shown in Response Fig. R11b. This is consistent with previous findings on other disordered solid electrolytes²³⁻²⁵. However, the overall trend remains consistent with previous results, and therefore, it does not affect any of our discussions or conclusions.

Response Fig. R11. (a) Mean squared displacement (MSD, $\overline{r^2(t)}$) of Li_3PS_4 glass at 300 K simulated for 10 ns. (b) Room temperature ionic conductivity obtained from long-time scale MD simulations at 300 K and extrapolation of Arrhenius fit.

Regarding the second part of the comment, we have now included additional discussions on the lag in the decrease of MSD with the increase in disorder as the temperature lowers. Specifically, it is due to the higher activation energy of diffusion in the crystal compared to that in glass and glass-ceramics (as further discussed in the response below), i.e., the migration rate of lithium ions decreases more rapidly as the temperature is lowered in the crystal. Owing to the disordered structure, lithium-ion migration in glass and glass-ceramic does not have preferred pathways, resulting in isotropic transport. The diverse energy landscape²⁶ provides more potential ion migration paths, allowing for a higher number of mobile ions even as the temperature decreases. In contrast, lithium migration in crystals follows preferred pathways¹⁸. As the temperature decreases, the ionic mobility weakens and lattice vibrations diminish, causing the migration channels to become more fixed, which significantly reduces the number of mobile ions. These additional discussions and figures have been included in the revised manuscript and Supplementary Information.

10. The discussion around Figures 2f and 2g presents a challenge in following the narrative coherently. There's a discrepancy between the temperatures mentioned in different sections; while some computations are conducted at very high temperatures (>700 K), others are performed at room temperature. This inconsistency makes it difficult to track the flow of the discussion. Additionally, the transition from discussing MSD and diffusion coefficients at high temperatures to ionic conductivities at room temperature, as depicted in Figure 2g, further complicates the understanding. The observation that the conductivity of the crystalline phase is higher than that of the glass and glass-ceramic phases in Figure 2g seems contradictory when considering the context of the high-temperature simulations discussed earlier. Moreover, the temperature range considered in Figure S8, spanning from 300 K to 700 K, lacks guidance or justification, adding to the confusion.

Response: As mentioned in our response to your Comment no. 9 above, we have simulated the lithium-ion diffusion from high temperatures down to room temperature and calculated the ionic conductivity. In the previous Fig. 2g (now Fig. 3b in the revised manuscript), we focus on comparing the room temperature conductivities of samples with different degrees of disorder, which is a critical indicator for solid-state electrolytes. Comparing the room temperature conductivity with experimental values is here possible and meaningful. In this regard, we need to point out that the room temperature ionic conductivity of the crystalline phase in our initial calculations is lower than that of the glass and glass-ceramic electrolytes. Even though the ionic conductivity of the crystal at high temperatures is higher than that of the glass and glass-

ceramic, its activation energy is relatively higher (i.e., related to a larger change in diffusion coefficient as a function of temperature). Therefore, whether extrapolated to room temperature or obtained from long-time scale simulations at room temperature, the ionic conductivity of the crystal is lower than that of the glass and glass-ceramic. We should also note that, as addressed in the response to comment no. 9, the MSD at 300 K does not reach the diffusion regime, i.e., the estimation of the diffusion coefficient is inaccurate at lower temperature.

The MSD curves in Figure S8 (now Fig. S13 of the revised Supplementary Information) are from different phases within the glass-ceramic, not used for Figures 2f and 2g. The temperature for MD simulation from 300 K to 1000 K is mainly for analyzing the ion dynamics. In the revised manuscript, we have reorganized the paragraphs for clearer logical presentation and discussion. We have also revised the figures and figure captions to prevent any misunderstanding.

11. Furthermore, the sudden shift in nomenclature in Figure 2e, from using $\beta/g/gc_subindex_temperature$ labels to referring directly to the β , glassy, and glass-ceramic phases, as consistently mentioned throughout the manuscript, adds to the difficulty in following the discussion seamlessly. Maintaining consistency in nomenclature throughout the figures and text would be desirable.

Response: Thank you for pointing this out. We have changed the abbreviations in the figures to be consistent throughout the text and figures of the revised manuscript and Supplementary Information.

12. At the beginning of page 8, the concept of “anomalous diffusion” is introduced without initial elaboration. It might be more straightforward to initially introduce the concept of dynamic heterogeneity, as it is later described and referenced from the literature. This approach would simplify the text and improve readability by avoiding unnecessary complexity. Additionally, the non-Gaussian parameter mentioned in line 184 on page 8 is not defined or described anywhere in the text, further complicating the understanding of the analysis. Without proper explanation, it becomes challenging to comprehend how from Figure 2b one can infer that “the motion of lithium ions exhibits non-Gaussian properties” as stated in line 188. While the position and height of the peak may seem relevant, readers are not informed about the significance or interpretation of these data points. Providing clear guidance on how to interpret these findings would enhance the flow of the discussion and make it easier to follow without needing to refer to the original paper.

Response: Following the Reviewer’s helpful suggestions, we have reorganized the sequence of concepts and modified the narrative logic to make it easier to understand in the revised manuscript. Specifically, we now first introduce dynamic heterogeneity and explain its non-Gaussian characteristics. Then, we define the non-Gaussian parameters and interpret the significance of the changes in these parameters over time.

Furthermore, we have also explained the relationship between the intensity and position of the peak and dynamic heterogeneity to make it easier to follow. In detail, dynamic heterogeneity is strongest when the NGP reaches its peak. For instance, some atoms or particles move very quickly (a behavior known as “hopping”), while others move relatively slowly. The position of the NGP peak (i.e., the time or displacement scale at which the peak occurs) is related to the timescale of non-Gaussian behavior in the system. For time-dependent NGP, the peak position is commonly used to characterize the timescales of different dynamic processes in the system, such as the glass transition and the relaxation of microstructures. At this moment, the displacement distribution of the system exhibits the greatest deviation from a Gaussian distribution.

13. The discussion on page 8, lines 205-207, regarding the orientation of the crystalline phases and the preferential diffusion path along the c-direction on the ac-plane in β -Li3PS4 appears disconnected from the observation that the glass-ceramic phase does not exhibit intermediate properties between the β and glass phases. This raises questions about the sensitivity of the results to the orientation of the ceramic-glass

system and why a specific interface orientation was chosen over others. Exploring alternative interface orientations and discussing their potential impact on the results could provide valuable insights into the robustness of the findings. Furthermore, the statement on page 8, lines 208-209, that “the z-direction is superior to that along the y-direction and significantly outperforms the x-direction” regarding diffusion requires more detailed discussion and analysis of the diffusion mechanisms involved. Simply describing the results without delving into the underlying mechanisms limits the depth of understanding.

Response: Thank you for your comment. To clarify, we have reorganized and added a more in-depth discussion in the revised manuscript. First, we discuss the priority of directions for lithium-ion diffusion in the crystal and its underlying mechanisms. Our finding is that lithium-ion diffusion along the c-axis is significantly better than that in other directions as shown in Response Fig. R12 (Supplementary Fig. S15 in the revised Supplementary Information). This is in agreement with related experimental and computational results^{1,18}.

Response Fig. R12. Lithium-ion hopping paths from different perspectives in the crystalline phase, where the lithium-ions and the selected lithium hopping trajectories are colored according to their depth in the b-direction.

Next, we address the sensitivity issue of crystal phase orientation in glass-ceramics. Specifically, crystals with different orientations in the glass-ceramics have been examined, as shown in Response Fig. R13, and they exhibit small differences. Since the enhancement in ionic conductivity is observed across all samples of glass-ceramic electrolytes (see Supplementary Fig. S11), including those with varying crystal content and orientations, we propose that the superior conductivity performance in glass-ceramics is driven by the disordered phase. Detailed discussions related to the above findings and the proposed mechanism are provided in the revised manuscript and Supplementary Information.

Response Fig. R13. (a) Temperature dependence of ionic conductivity of glass-ceramic Li_3PS_4 . Three different orientations of crystalline phases within the glass-ceramic are considered. (b,c) MSD of glass-ceramic Li_3PS_4 at (b) 300 K and (c) 900 K.

14. The discussion on page 9, spanning lines 247 to 252, presents a challenge in readability as it involves referencing and comparing five different graphs from both the main manuscript and supplementary information. This dense concentration of information makes it difficult for readers to follow the discussion seamlessly. To enhance readability, it may be beneficial to streamline the discussion by consolidating the comparisons and providing concise summaries of the key findings from each graph. Perhaps, including visual aids such as tables summarizing the relevant data could assist readers in comprehending the comparisons without needing to constantly refer back and forth between multiple figures.

Response: Thank you for the suggestions. We have reorganized the logic of the relevant sections. Before each discussion, we have provided concise summaries of the main findings for each figure. Additionally, we have reformatted the figures, consolidating similar results under the same discussion into one figure, thereby reducing the need for back-and-forth referencing of figures during the discussion. Finally, we have summarized the discussed results in a single figure, as shown in Response Fig. R14 (reproduced as Supplementary Fig. S14), making the correspondence between the discussion and results more intuitive and easier to understand.

Response Fig. R14. The non-Gaussian parameter (NGP) peak times, τ_{ngp} for crystalline, interface, and glass phases in glass-ceramic Li_3PS_4 (hollow symbols) The solid symbols provide the value of τ_{ngp} for glassy Li_3PS_4 , $\beta\text{-Li}_3\text{PS}_4$, and glass-ceramic Li_3PS_4 electrolytes.

15. The final discussion provided on page 13 (lines 351-364) feels somewhat inadequate given the extensive volume of data presented throughout the paper. This raises questions about the suitability of the journal format for effectively conveying such complex findings. Moreover, the apparent conclusions drawn from the discussion lack generalization, leaving gaps in understanding. With the complexity of the analyses, it remains unclear how disorder in the glass-ceramic system enhances ionic conductivity at low temperatures. It's possible that this enhancement is merely a consequence of the intrinsic lower activation energy of the present disordered phase. The discussion would benefit from further exploration of the role of Li-ion diffusion through the interface between the glass and crystalline phases in the glass-ceramic model. If the interface is thought to block Li-ion diffusion, then how is percolation of Li-ion observed in the study? This discrepancy is puzzling and warrants a more thorough examination and explanation. Overall, a deeper analysis and synthesis of the findings are needed to provide a more comprehensive and generalizable conclusion.

Response: We agree with the Reviewer and have added more discussion in the revised manuscript, explaining the mechanisms by which disorder enhances ion conduction and the role of interfaces. Specifically, we find that dynamic heterogeneity plays a key role in controlling lithium diffusion since the dynamic heterogeneity can be frozen-in upon cooling, leading to static or structural heterogeneity that impacts the room-temperature ionic conductivity, and it has been demonstrated that lithium-ion percolation between different phases in the glass-ceramics leads to cooperative diffusion, driven by the disordered phase. Additionally, we have provided a more comprehensive conclusion.

In Fig. 4 of the revised manuscript (Response Fig. R15 below), we further confirm that the more disordered interfacial phase and glassy phase facilitate the cooperative migration of lithium ions in the crystalline phase and do not block lithium diffusion. This is why lithium percolation between different phases as shown in Fig. 4d of the revised manuscript.

Response Fig. R15. Diffusion dynamics from order to disorder. **a** Time-averaged mean-squared displacement (MSD) and **b** non-Gaussian parameter (NGP) of lithium ions for glass-ceramic Li_3PS_4 system. **c** Self-part van Hove correlation function for glass-ceramic Li_3PS_4 system. **d** Time profiles of lithium-ion fraction within each phase of the glass-ceramic Li_3PS_4 system. **e** Atomic snapshots of lithium ions migration trajectories within glass (left), crystalline (center), and interfacial (right) phases of the glass-ceramic Li_3PS_4 system over a timespan of 1 ns. **f** Short-term MSD of crystalline phase in glass-ceramic Li_3PS_4 . **g** Distinct-part van Hove correlation function for $\beta\text{-Li}_3\text{PS}_4$ (top) and crystalline phase in glass-ceramic Li_3PS_4 (bottom). **h** Schematic diagram of lithium-ion hopping paths from different perspectives in the crystalline phase, where the lithium ions and the selected lithium hopping trajectories are colored according to their depth in the b -direction. (reproduced as Fig. 4 from the revised manuscript).

16. In the methods section, the choice of performing AIMD simulations at very high 3000 K raises concerns about its representativeness of actual glassy structures. The authors should provide justification for this procedure and compare it with alternative approaches. It's essential to ensure that the sampling is sufficiently robust, universal, and representative of the actual energy landscapes encountered in the formation of glassy

or disordered structures. The selection of such a high temperature and short MD run duration (2.5 ps) may lead to a bias towards exploring configurations with very high energies, potentially deviating from the energy landscape of actual glassy structures. There's a risk of overlooking important configurations and dynamics that are crucial for understanding the formation of glassy or disordered structures. Therefore, the authors should thoroughly justify their choice of simulation parameters and demonstrate how they ensure the sampling captures the relevant energy landscape effectively.

Response: First, we acknowledge that the high temperature simulation does not represent the real glass structure, which should be generated from the simulation of cooling the melt at a reasonable cooling rate. However, the typically simulated quenching rate on the order of 10^{12} K/s is much higher than the experiment value of 1-100 K/s, which leads to a more disordered structure in the simulation (Li et al., J. Chem. Phys. 147, 074501 (2017)). However, compared to the structure difference between glasses quenched by varying cooling rates, crystals exhibit a significantly different structure than glasses. Therefore, the main conclusions will not be influenced by this discrepancy as the focus of this study is glass-ceramics, which relies on the different combinations of glass and crystal.

Second, we would like to point out that the datasets for training the MLIP do not solely rely on the AIMD simulations at 3000 K as explained in the following. However, the motivation of selecting a high temperature is to explore the energy landscape as much as possible within the accessible time limited by the expensive AIMD simulation. This is important as the melt-quench simulation to form glass will explore a large region within the energy landscape, which has been widely applied in building the initial datasets for MLIP training in an efficient manner (e.g., see Musaelian et al. Nat. Commun. 14, 579 (2023) and Du et al. Natl. Sci. Rev. 11, nwae023 (2024)). As pointed out by the Reviewer, we have indeed noticed that the high temperature AIMD simulation may bias the sampling of high energy phases. Therefore, we have also added more datasets at relatively lower temperatures (400~1800 K) combined with different pressures using the active learning protocol. The detailed information of the initial datasets and explored datasets are provided in Supplementary Tables S2 and S3 of the revised Supplementary Information. These two datasets cover the different compositions and phases of LPS systems under various environments for training the MLIP.

Furthermore, as the initial datasets mainly deal with the melting state of LPS, the explored datasets under different combinations of temperature and pressure cover the majority conditions of LPS systems in the viscous and solid states. The combination of two datasets allows us to capture the commonly explored energy landscape of LPS systems, and the resulting MLIP shows a good generatability generalization ability. For example, it can well predict the interfacial structure of β -Li₃PS₄ that was not included in the training sets (see response to Comment no. 4 of Reviewer 1 above). Finally, as discussed in relation to Comment no. 1 of Reviewer 1, the completeness of the datasets can be further confirmed by the finding that the present MLIP can accurately predict the interaction within the α -Li₃PS₄ phase, while the accuracy cannot be significantly increased when including α -Li₃PS₄ (see Response Fig. R1).

17. The validation of the MLIP presented in Figure S1, particularly regarding the neutron structure factor, is not entirely satisfactory. There appears to be a discrepancy in both the position and relative intensities of the two main peaks in the range between 0 and 2.5 Q when compared to experimental data. Specifically, the simulated data is right-shifted with respect to the experiment, and the relative intensities of the peaks do not align as expected. Moreover, in Fig 1a, it's observed that the agreement between the MLIP and AIMD starts to deviate in the medium to long range, particularly beyond 3 Angstroms. In many cases, MLIPs are designed to capture short-range interactions accurately, as these typically dominate atomic-scale processes such as bonding and structure. Therefore, minor discrepancies in the medium to long range may not necessarily be alarming, especially if the MLIP accurately reproduces the essential features of the system at shorter distances. However, if the glassy structure is sensitive to long-range interactions, or if the deviations in the medium to long range become more pronounced with increasing distance, then this discrepancy may warrant

further investigation. It's essential to assess the impact of these deviations on the overall accuracy and predictive capability of the MLIP.

Response: In the beginning of the Results section in the revised manuscript, we have added a new subsection entitled "Machine learning interatomic potential". This includes more discussion on the validation of the accuracy of the machine learning potential. Furthermore, we would like to refer the Reviewer to our related responses to Reviewer #1's Comments no. 1-4 and 6-7 above. As shown in Response Fig. R5 (Supplementary Fig. S1 of the revised Supplementary Information), the calculated structure factor of Li_3PS_4 glass using the MLIP features a very good agreement with the experimental results.

18. In the methods section, it would be beneficial to provide commentary on the level of convergence achieved by the employed basis sets. This information is important for ensuring the reliability and accuracy of the results obtained from the simulations.

Response: Good point. Before the DFT calculations, we had performed the convergence test on the employed basis sets. The Quickstep module of the CP2K package uses a multi-grid system for mapping the electron density and the product Gaussians onto the real-space grid. The accuracy of the DFT calculations thus heavily relies on the grid size, which is defined by the plane-wave cutoff (cutoff) and the relative cutoff at which a Gaussian is mapped (Rel_cutoff). These two cutoff values should be high enough to achieve an accurate calculation. To this end, we have systematically investigated the dependence of total energy of $\beta\text{-Li}_3\text{PS}_4$ on these two values. As shown in Response Fig. R16, the plane-wave cutoff of 500 Ry and relative cutoff of 50 Ry can ensure the accuracy of the DFT calculations. All the related discussions and figures have been included in the revised manuscript and Supplementary Information.

Response Fig. R16. The dependence of total energy of $\beta\text{-Li}_3\text{PS}_4$ on (a) the plane-wave cutoff for the electronic density and (b) the relative cutoff. The values highlighted in red circles are adopted in this study.

19. And again, in the methods section, it is relevant to include information on dataset size convergence during the training procedure. Ensuring dataset size convergence provides confidence that the diversity of the dataset structures has been adequately captured. This convergence assessment could involve monitoring the changes in model performance metrics, such as accuracy or error, as the dataset size increases.

Response: We agree that the size of the dataset should be large enough to cover the possible combinations of compositions and structures. Due to the generalization ability of deep learning, our MLIP can be applied in the simulation of various LiPS phases, given the atomic interactions described by MLIP are based on the local environments of individual atoms. Therefore, as explained in details in our responses to the comments of Reviewer #1, the present MLIP can well predict the interactions within $\alpha\text{-Li}_3\text{PS}_4$ crystal (comment no. 1) and

the crystal-amorphous interfaces of β - Li_3PS_4 (comment no. 4), although these two structures were not explicitly included in the training datasets. The convergence of the datasets is also confirmed from these findings since a new MLIP trained with these structures cannot further improve its accuracy. We have provided this information in the Methods section of the revised manuscript.

20. How have been obtained the number of nearest neighbors per atom in Equation 7? Using Voronoi partition perhaps? Please clarify. And please, clarify as well if the redefinition of Equation 7 as Equation 8 aims to make the Steinhardt order parameter rotationally invariant or it has other motivations. Perhaps it's also worth considering alternative variants of Steinhardt order parameters, such as weighted versions [<https://doi.org/10.1063/1.4774084>] or w_l indexes [<https://doi.org/10.1103/PhysRevB.28.784>]. These variants may offer additional insights or advantages.

Response: Yes, we have used Voronoi and weighted Steinhardt order parameter according to the Voronoi tessellation, and the number of nearest neighbors was obtained from the Voronoi tessellation. We have also incorporated the variant described in the literature that the Reviewer refers to, using w_l order parameters for comparison. Notably, we obtained the same conclusions as shown in Fig. S21 of the revised Supplementary Information. Response Fig. R17 compares the q_l and w_l (Steinhardt order parameter and the variant w_l order parameters). We have added more detailed descriptions in the Methods section in the revised manuscript and Supplementary Information.

Response Fig. R17. Profiles of the averaged Steinhardt order parameter (q_l) and w_l order parameters along the y -direction in the glass-ceramic Li_3PS_4 configuration.

Other minor issues:

21. In the caption of Figure 1e, for consistency, it would be helpful to state that the solid line represents the RDFs and the dashed line represents the integrated RDF.

Response: Thank you for the careful reading. In the figure caption of the revised manuscript, we have now stated that the solid and dashed lines represent different functions.

22. In page 6, line 162, referring to Figs. S5d-e appears to be a typographical error. It should likely refer to Figs. S5a-c instead.

Response: Correct, thank you. We have revised the order of figure for correct numbering.

23. Regarding the discussion of Figure S5 on page 6, it may be clearer to combine the data for the three systems (beta, glass, glass-ceramic) into a single figure with the three curves at 700 K. This would provide a more direct comparison and facilitate understanding.

Response: As suggested, we have combined them into a single figure.

24. The considered temperature in Figure 4a and S11 should be explicitly stated in the figure captions or the surrounding text to ensure clarity.

Response: We have specified the temperature at which the data was calculated in the figure captions.

25. The concept of "Voronoi volume of lithium," introduced in the caption of Figure S4, should be described in the text.

Response: We have added a description and discussion of the Voronoi volume in the revised manuscript.

26. The fraction of crystallinity presented in Figure S10 should be introduced and defined more smoothly in the main text.

Response: Thank you again. We have defined the fraction of crystallinity in the revised manuscript.

Response References:

1. Homma, K. *et al.* Crystal structure and phase transitions of the lithium ionic conductor Li_3PS_4 . *Solid State Ion.* **182**, 53–58 (2011).
2. Tachez, M., Malugani, J.-P., Mercier, R. & Robert, G. Ionic conductivity of and phase transition in lithium thiophosphate Li_3PS_4 . *Solid State Ion.* **14**, 181–185 (1984).
3. Chen, Y. *et al.* Correlation of anisotropy and directional conduction in $\beta\text{-Li}_3\text{PS}_4$ fast Li^+ conductor. *Appl. Phys. Lett.* **107**, 013904 (2015).
4. Holekevi Chandrappa, M. L., Qi, J., Chen, C., Banerjee, S. & Ong, S. P. Thermodynamics and Kinetics of the Cathode–Electrolyte Interface in All-Solid-State Li–S Batteries. *J. Am. Chem. Soc.* (2022) doi:10.1021/jacs.2c07482.
5. Žguncs, P. & Yildiz, B. Strain Sensitivity of Li-ion Conductivity in $\beta\text{-Li}_3\text{PS}_4$ Solid Electrolyte. *PRX Energy* **1**, 023003 (2022).
6. Gigli, L., Tisi, D., Grasselli, F. & Ceriotti, M. Mechanism of Charge Transport in Lithium Thiophosphate. *Chem. Mater.* **36**, 1482–1496 (2024).
7. Ariga, S., Ohkubo, T., Urata, S., Imamura, Y. & Taniguchi, T. A new universal force-field for the $\text{Li}_2\text{S}\text{-P}_2\text{S}_5$ system. *Phys. Chem. Chem. Phys.* **24**, 2567–2581 (2022).
8. Jalem, R., Chandrappa, M. L. H., Qi, J., Tateyama, Y. & Ong, S. P. Lithium dynamics at grain boundaries of $\beta\text{-Li}_3\text{PS}_4$ solid electrolyte. *Energy Adv.* **2**, 2029–2041 (2023).
9. Hayashi, A., Hama, S., Minami, T. & Tatsumisago, M. Formation of superionic crystals from mechanically milled $\text{Li}_2\text{S}\text{-P}_2\text{S}_5$ glasses. *Electrochem. Commun.* **5**, 111–114 (2003).

10. Zhou, R., Luo, K., Martin, S. W. & An, Q. Insights into Lithium Sulfide Glass Electrolyte Structures and Ionic Conductivity via Machine Learning Force Field Simulations. *ACS Appl. Mater. Interfaces* **16**, 18874–18887 (2024).
11. Ohara, K. *et al.* Structural and electronic features of binary Li₂S–P₂S₅ glasses. *Sci. Rep.* **6**, 21302 (2016).
12. Cubuk, E. D. *et al.* Identifying Structural Flow Defects in Disordered Solids Using Machine-Learning Methods. *Phys. Rev. Lett.* **114**, 108001 (2015).
13. Schoenholz, S. S., Cubuk, E. D., Sussman, D. M., Kaxiras, E. & Liu, A. J. A structural approach to relaxation in glassy liquids. *Nat. Phys.* **12**, 469–471 (2016).
14. Huang, J. *et al.* Deep potential generation scheme and simulation protocol for the Li₁₀GeP₂S₁₂-type superionic conductors. *J. Chem. Phys.* **154**, 094703 (2021).
15. Bonati, L. *et al.* The role of dynamics in heterogeneous catalysis: Surface diffusivity and N₂ decomposition on Fe(111). *Proc. Natl. Acad. Sci.* **120**, e2313023120 (2023).
16. Dietrich, C. *et al.* Lithium ion conductivity in Li₂S–P₂S₅ glasses – building units and local structure evolution during the crystallization of superionic conductors Li₃PS₄, Li₇P₃S₁₁ and Li₄P₂S₇. *J. Mater. Chem. A* **5**, 18111–18119 (2017).
17. Preefer, M. B. *et al.* Subtle Local Structural Details Influence Ion Transport in Glassy Li⁺ Thiophosphate Solid Electrolytes. *ACS Appl. Mater. Interfaces* **13**, 57567–57575 (2021).
18. Kaup, K., Zhou, L., Huq, A. & Nazar, L. F. Impact of the Li substructure on the diffusion pathways in alpha and beta Li₃PS₄: an in situ high temperature neutron diffraction study. *J. Mater. Chem. A* **8**, 12446–12456 (2020).
19. Chandra, A., Bhatt, A. & Chandra, A. Ion Conduction in Superionic Glassy Electrolytes: An Overview. *J. Mater. Sci. Technol.* **29**, 193–208 (2013).
20. Famprikis, T., Canepa, P., Dawson, J. A., Islam, M. S. & Masquelier, C. Fundamentals of inorganic solid-state electrolytes for batteries. *Nat. Mater.* **18**, 1278–1291 (2019).
21. Stegmaier, S. *et al.* Nano-Scale Complexions Facilitate Li Dendrite-Free Operation in LATP Solid-State Electrolyte. *Adv. Energy Mater.* **11**, 2100707 (2021).
22. *Springer Handbook of Glass*. (Springer International Publishing, Cham, 2019). doi:10.1007/978-3-319-93728-1.
23. Raiteri, P., Demichelis, R. & Gale, J. D. Thermodynamically Consistent Force Field for Molecular Dynamics Simulations of Alkaline-Earth Carbonates and Their Aqueous Speciation. *J. Phys. Chem. C* **119**, 24447–24458 (2015).
24. Xu, W.-S., Douglas, J. F. & Freed, K. F. Influence of Cohesive Energy on Relaxation in a Model Glass-Forming Polymer Melt. *Macromolecules* **49**, 8355–8370 (2016).
25. Zhu, Y. *et al.* Highly disordered amorphous Li-battery electrolytes. *Matter* **7**, 500–522 (2024).
26. Dyre, J. C., Maass, P., Roling, B. & Sidebottom, D. L. Fundamental questions relating to ion conduction in disordered solids. *Rep. Prog. Phys.* **72**, 046501 (2009).